# Improved RNA stability estimation indicates that transcriptional interference is frequent in diverse bacteria

Walja C. Wanney [1,2,3], Loubna Youssar[1,3], Gergana Kostova[1] & Jens Georg [1✉]

We used stochastic simulations and experimental data from *E. coli*, *K. aerogenes*, *Synechococcus* PCC 7002 and *Synechocystis* PCC 6803 to provide evidence that transcriptional interference via the collision mechanism is likely a prevalent mechanism for bacterial gene regulation. Rifampicin time-series data can be used to globally monitor and quantify collision between sense and antisense transcription-complexes. Our findings also highlight that transcriptional events, such as differential RNA decay, partial termination, and internal transcriptional start sites often deviate from gene annotations. Consequently, within a single gene annotation, there exist transcript segments with varying half-lives and transcriptional properties. To address these complexities, we introduce 'rifi', an R-package that analyzes transcriptomic data from rifampicin time series. 'rifi' employs a dynamic programming-based segmentation approach to identify individual transcripts, enabling accurate assessment of RNA stability and detection of diverse transcriptional events.

[1] Genetics and Experimental Bioinformatics, Faculty of Biology, Albert-Ludwigs-Universität Freiburg, Freiburg, Germany. [2] Present address: Plant Biotechnology, Faculty of Biology, Albert-Ludwigs-Universität Freiburg, Freiburg, Germany. [3] These authors contributed equally: Walja C. Wanney, Loubna Youssar. ✉email: jens.georg@biologie.uni-freiburg.de

Genes in prokaryotes are often organized into polycistronic operons, which combine several genes under the control of a single promoter. Overall, this arrangement harmonizes gene expression, but it lacks the needed fine-tuning for different RNA abundances of individual genes within operons[1]. Internal transcription start sites (iTSS) and partial termination can divide the operon into distinct units[2]. Another mechanism is the differential operon decay caused by stabilization sites in the RNA structure[3]. Post-transcriptional gene regulation can also be achieved by the binding of *trans*-acting sRNAs[4] or *cis*-antisense RNAs (asRNA) to the nascent transcripts[5–7]. An interesting mechanism is the steric interaction of RNA polymerases (RNAPs), a mechanism known as transcription interference (TI)[8]. TI has been well analyzed on a theoretical level and with synthetic constructs[9–12]. However, only few reports exist on naturally occurring TI[13–17] and it is unknown how widespread TI actually is in bacteria. We employed stochastic simulations and developed strategies to detect these patterns in high-resolution rifampicin datasets from enterobacteria, *E. coli*[3,18] and *Klebsiella aerogenes* KCTC 2190[3] and Cyanobacteria, *Synechococcus* PCC 7002[19] and *Synechocystis* PCC 6803. Our approach ('rifi') allows inference of the in vivo termination effect of TI and other factors influencing transcription termination, such as *trans*-acting sRNAs. We provide several examples of TI facilitated by asRNAs, including the *nuo* or *mazEF* operons from *E. coli*. Moreover, we demonstrate instances of TI between 3′−3′ overlapping genes and transcription termination mediated by sRNAs (Spot42, RybB, DicF).

Rifampicin, an antibiotic commonly used to study RNA half-lives in bacteria, inhibits transcription initiation but does not affect elongating RNAPs[20,21]. Chen et al.[22] observed that this property can lead to a delay of the exponential decay of RNA in which the RNA concentration remains constant. The delay depends on the time that the last elongating polymerase needs to pass the investigated transcript position and increases with the distance to the transcriptional start site (TSS). Furthermore, different modes of RNA decay have been hypothesized[22]. For the "co-transcriptional decay" it is assumed that the decay is already possible during RNA synthesis, e.g. by 5′ exo- or endonucleases. In contrast, "post-transcriptional decay" means that the decay is only possible from the 3′ end of the full-length transcript[22]. We show that the different decay modes lead to very different RNA abundance patterns that allow to differentiate between them.

Our approach 'rifi' differs from previous studies analyzing rifampicin data[3,18,19,22,23], because it is not relying on a given genome annotation for, e.g., the calculation of transcript half-lives. Instead, we used dynamic programming to combine bins (RNA-Seq) or probes (microarray) of similar behavior. That allows to detect transcript segments with different properties within one and the same annotated gene. Around 90 *E. coli* genes are associated with two or more transcripts with half-lives that differ by factors from 1.7 to 14. Using 'rifi', an R-package designed for platform- and organism-independent holistic analysis of high-resolution rifampicin time series data, we obtained the following findings: (1) We provide evidence for frequent gene regulation through transcriptional interference in bacterial model organisms. (2) The RNA decay can initiate already during transcription (co-transcriptional decay) in the investigated bacteria. (3) The mode of decay can be inferred from abundance patterns in standard transcriptomics data. (4) The transcriptome often deviates from genome annotation, highlighting the benefits of annotation-independent analysis for more accurate half-life calculations. (5) It is possible to extract transcriptional features such as termination sites and efficiency, elongation rates, TSS locations, and operon structures from rifampicin data.

## Results and discussion

**Decay pattern analysis reveals increased abundance for certain transcripts upon rifampicin addition.** The initial objective was to estimate global half-lives of transcripts in the cyanobacterium *Synechocystis* PCC 6803 by analyzing time-resolved microarray transcriptome data after rifampicin treatment. Surprisingly, soft-clustering of the time-series data from individual probes identified clusters with an initial abundance increase before onset of the exponential decay (Fig. 1). To determine if this pattern was present in other rifampicin time-series data we investigated four datasets from *E. coli* BW25131[3], *E. coli* MG1655[18], *Klebsiella aerogenes* KCTC 2190[3] and *Synechococcus* PCC 7002[19]. The increase was detectable in all datasets (Fig. 1), indicating that this phenomenon is not specific to *Synechocystis* or our experimental workflow. In the following we propose biological explanations for the phenomenon and investigate the possibility of technical artifacts.

**Decay modes and decay scenarios.** We built a stochastic simulation framework to follow the synthesis and degradation of individual transcripts and verified that the simulation results actually followed the expected curves for the co-transcriptional decay (Supplementary Fig. 1). Next, we modified the initially published model for the post-transcriptional decay[22], which was not able to describe the simulation results (Supplementary Fig. 2). The decay mode has a drastic effect on the RNA accumulation during steady-state expression prior to rifampicin addition (Fig. 2). In case of co-transcriptional decay, RNA concentrations remain constant throughout the entire length of the transcript (Fig. 2a). Conversely, in the hypothetical post-transcriptional mode, RNA concentrations vary depending on the position within the transcript (Fig. 2b). The concentrations are highest closer to the 5′ end, while the very 3′ end has the steady-state concentration. This discrepancy arises because the 5′ end is present in all transcripts, while the very 3′ end is exclusive to the full-length transcript (Fig. 2b). Distinguishing between decay modes based on isolated decay curves is not possible (Supplementary Fig. 2). Instead, we examined consecutive bins within a single transcript. For post-transcriptional decay, there should be a linear decrease in positional RNA concentrations before rifampicin treatment. Additionally, the decay curves of all positions within the same transcript must align on a common exponential decay curve (Fig. 2b, Supplementary Fig. 2). For short transcripts and high elongation rates, both decay modes exhibit similar decay and abundance patterns, making them indistinguishable (Supplementary Fig. 3). Investigation on the four organisms did not yield evidence for the post-transcriptional decay mode in longer transcripts. The majority of transcripts did not exhibit a significant position-dependent decrease in RNA abundance. While certain transcripts, such as the *E. coli gap* pseudogene or the 3′ portion of the *Synechocystis sll1951* gene, showed a positional decline pattern, the decay curves of consecutive positions did not align as expected for post-transcriptional decay. Moreover, the position-dependent abundance decrease followed an exponential curve rather than a linear one (Supplementary Fig. 4). This effect could be attributed to a fixed termination probability after each elongation step, possibly resulting from the collision mode of transcription interference (TI) or a higher random termination rate of untranslated RNAs, as reported in *E. coli*[10,24]. The simplest mechanisms for co-transcriptional decay involve a 5′ exonucleolytic decay, similar to what is observed in *B. subtilis*. Both *Synechocystis* and *Synechococcus* possess a homolog of *B. subtilis* RNase J1, with 5′ exoribonuclease activity[19,25,26]. In *E. coli*, RNA decay primarily occurs through the combined action of endo- and

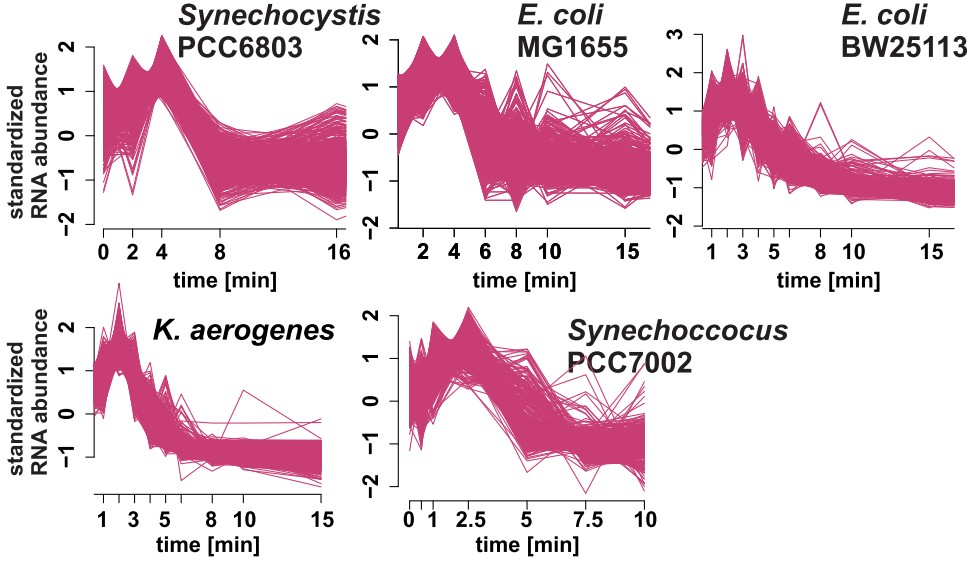

**Fig. 1 Clusters of decay patterns from *Synechocystis* PCC 6803, *E. coli*, *Klebsiella* and *Synechococcus* PCC 7002 that show an RNA abundance increase after rifampicin addition.** Individual probes (microarray) or bins (RNA-Seq) were separated into 20 clusters per dataset based on soft-clustering of standardized expression levels in a time course after rifampicin addition. The figure shows one representative cluster for each organism that shows the abundance increase pattern.

3′ exoribonucleases[27]. A simulation example demonstrates that a small number of internal processing sites coupled with rapid 3′ exonucleolytic decay can mimic co-transcriptional decay mediated by a 5′ exoribonuclease (Supplementary Fig. 5). Recent studies discovered a 5′ exoribonuclease, RNase AM, also in *E. coli*[28,29].

**Explanations for a post-rifampicin abundance increase**. We explored two biological scenarios to explain the increase in RNA abundance after rifampicin treatment:

1. *Pre-steady-state case*: The transcription of the transcript began shortly before the addition of rifampicin, and the concentration at the given position had not yet reached a steady state (Fig. 2a, Supplementary Fig. 6a). In this case, the concentration would increase from t = 0 until either the last elongating RNA polymerase passes this position or the steady state is achieved.

2. *Rifampicin sensitive termination (RST)*: A partial termination event occurs upstream of the investigated position, causing not all RNAPs that initiated at the TSS to reach this position. However, upon rifampicin addition, this termination is relieved, allowing elongating polymerases upstream of the termination site to reach the investigated position (Fig. 3, Supplementary Fig. 6b–e). Importantly, this scenario relies on the termination being sensitive to rifampicin. Since rifampicin targets RNAPs, it is likely that the termination process depends on a short-lived or non-processive factor, such as an sRNA or a protein that require constant resynthesis. However, the simplest explanation is that transcription interference (TI) by the collision mechanism[8] is responsible for the termination. In collision interference, RNAPs transcribing in opposite directions collide, resulting in the termination of one or both transcription processes. This requires continuous de novo transcription initiation from the interfering promoters. If there is an asymmetry in the time window during which elongating polymerases are present for one but not the other of the interfering transcripts the termination by collision will be stopped for the former transcript. Consequently, a wave of relatively higher RNAP concentration per nucleotide will pass the post-termination positions (Fig. 3).

Both scenarios can account for the post-rifampicin abundance increase. Example simulations and corresponding model fits are presented in Supplementary Fig. 6 and real data examples in Fig. 4. The two scenarios can be differentiated based on the intensities observed at different positions within a transcript before rifampicin addition (Supplementary Fig. 6). Technical artifacts such as sequencing noise (Fig. 5a, Supplementary Fig. 7) and a normalization artifact (Supplementary Fig. 8) were excluded as reason for the post-rifampicin abundance increase (details next to Supplementary Fig. 7).

**Rifampicin sensitive termination: examples for sRNAs and transcriptional interference**. In total, we observed 66 to 327 RST (Supplementary Data 1) segments across the studied organisms (Fig. 5b). Among them, 38 to 269 segments exhibited a distinct pre- and post-RST phase. Additionally, 23 to 177 RST events were linked to identified termination events, characterized by a sharp decline in the synthesis rate (further details below), indicating an impact on the expression of the corresponding transcript. Presence of a clear RST boundary and association with a termination event distinguishes RST from cases of "pre-steady-state" expression. The differences between the RST numbers in the investigated organism are most likely due to the different time-resolutions in the experimental setups as discussed below. Table 1 provides 16 selected RST instances from *E. coli* BW25113, including example RST events at the junctions of *mazE/F* (Fig. 4a), *dam/rpe* (Supplementary Fig. 9), and *nuoC/E* (Fig. 4c), which are correlated with detectable or reported asRNAs.

One of the detected RST events correlates with the position of the asRNA to the *mazEF* toxin/antitoxin transcript (aMEF). Gundy et al. [7] demonstrated that absence of aMEF results in an increase in polycistronic *mazEF* transcript and a decrease in the shorter *mazE* transcript, aligning with the ~34% partial termination observed through RST analysis (Fig. 4a). Wang et al. showed that binding of the sRNA Spot42 enhances termination at the *galT/K* junction[30], which corresponds to an identified RST site (Fig. 4b). We also see RST at the known *yebK/pykA* DicF[31] and *rbsC/K* RybB[32] sRNA binding sites which have not previously been associated with sRNA-dependent termination (Supplementary Data 2). Additionally, 14 potential instances of RST are

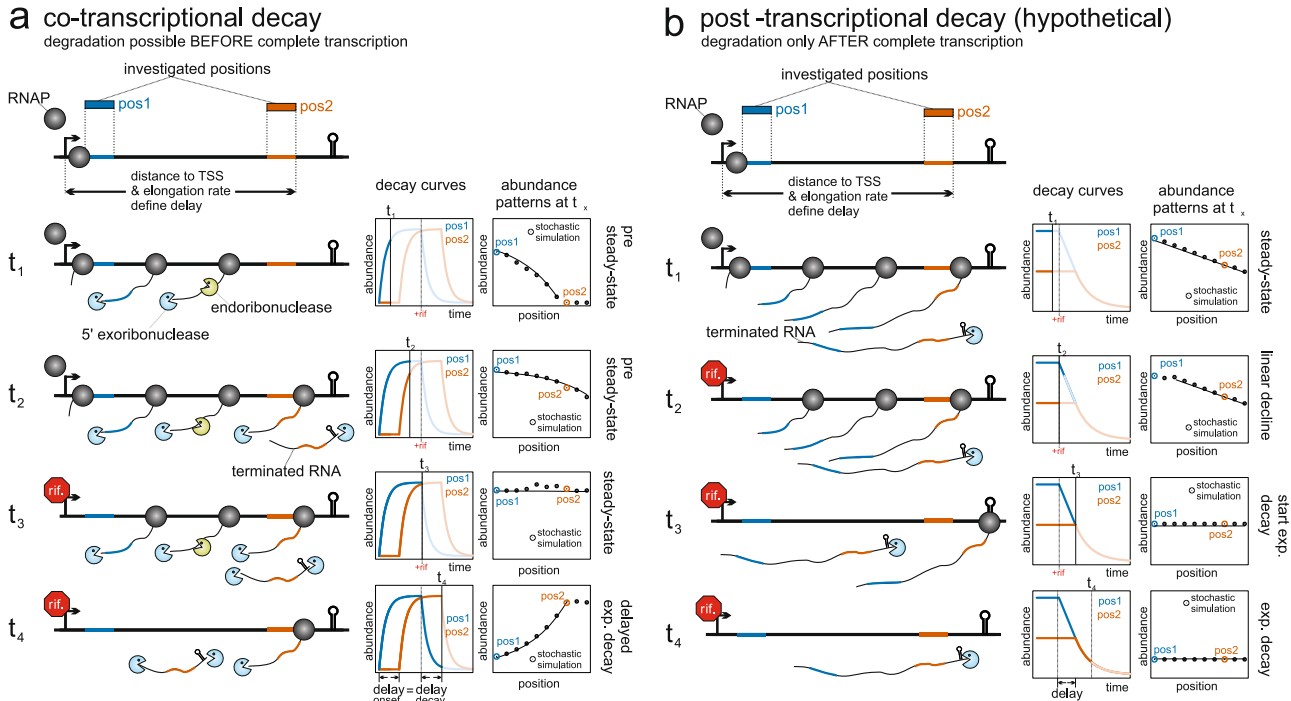

**Fig. 2 Comparison of different decay scenarios.** Two positions are highlighted, one directly behind the TSS (pos1, blue) and another at the end of the transcript (pos2, orange). For all considered time points ($t_x$) the decay curves for the two positions (decay curves, left column of plots) and the RNA concentrations at $t_x$ are shown as a function of the distance to the TSS (abundance patterns at $t_x$, right column). **a** *Co-transcriptional decay*: $t_1$ - the transcription of the gene has just started. The first RNAP has passed pos1 but is not yet at pos2, i.e. the RNA concentration at pos1 is higher than at pos2. The expression has not reached an equilibrium between synthesis and degradation rate, i.e. the steady state has not yet reached and the RNA concentrations increase over time. The time difference, the delay, between the onset of transcription at pos1 and pos2 depends on the elongation rate and the distance between the positions and the TSS. The abundance patterns show a position-specific decline of the RNA concentrations. $t_2$—the concentration at pos1 but not at pos2 has reached its steady state. The positional abundance decline is shallower at this later time point. $t_3$—if rifampicin is added, after the steady state for all positions is reached, all positions have the same concentration. Due to the stochastic co-transcriptional decay by 5′ exo- and endonucleases, no transcript position is overrepresented. The RNA concentration at a given position stays constant, i.e. the exponential decay is delayed until the last elongating RNAP has passed it. $t_4$—the last RNAP has passed pos2 and the exponential decay now starts at this position as well. **b** *Post-transcriptional decay*: $t_1$—(steady state) if only full-length transcripts can be degraded by 3′ exonucleases, the 5′ end is more frequent than downstream positions. This leads to a position-specific decline in the abundance patterns in the steady state. $t_2$—after addition of rifampicin the concentration stays constant until the last polymerase has passed the respective position. This is followed by a linear decline which represents the remaining transcription of the full-length transcript from all RNAPs that are still elongating. $t_3 + t_4$—after the last RNAP has reached the 3′ end, the exponential decay becomes visible. Note that, in contrast to the co-transcriptional decay, the linear decline and the exponential decay fall on the same curve for all transcript positions.

conserved in the data of the closely related strains *E. coli* BW25113, *E. coli* MG1655 and/or *K. aerogenes* KCTC 2190 (Table 1). An example is the RST event in the *E. coli* *nuoABCEFGHIJKLMN* operon which is also conserved in *Klebsiella* (Fig. 4c). Other examples are RST instances between *dam* and *rpe*, (Supplementary Fig. 9) and at the *mazE/F* junction which are present in the two *E. coli* strains.

**Excludons and overlapping mRNA-mRNA transcription.** Also protein-coding transcripts can have sense-antisense overlapping transcription[2,33]. These overlaps can have a regulatory function, e.g. if the gene products of the opposing transcripts also have antagonistic functions, then called excludon[34]. While it is relatively easy to detect regions of overlapping coverage in transcriptome data, it is not directly evident if this overlap actually results in e.g., co-degradation or TI. In an example from *Synechococcus* there is a 3′−3′ overlap between a TU with 7 genes and a single gene TU (Fig. 4d). The 3′ bins of the shorter transcript showed no indication of RST, but a clear position-dependent decline of the synthesis rate. For the 3′ bins of the long TU a clear position-dependent increase in the RST rate is accompanied with a decrease in the synthesis rate. For bins at the

very 3′ end of the operon the increase is delayed, which distinguishes RST from pre-steady-state expression. A potential excludon from *E. coli* and *Klebsiella* is the 3′−3′ overlapping *matP/ompA* pair (Table 1, Supplementary Data 2). In *E. coli*, *K. aerogenes* and *Synechocystis* only a small number of RST events in 3′−3′ overlapping transcripts were detected. However, in *Synechococcus* 107/177 RST events belong to this class (Supplementary Data 1). It is unclear if this high number has technical reasons or if 3′−3′ overlapping transcription is a more general termination mechanism in *Synechococcus*.

**Potential transcriptional interference at sites with no detectable asRNA coverage.** Many RST instances do not correlate with a detectable asRNA or a known sRNA binding site. An interesting example is the asRNA to the *mazEF* toxin/antitoxin transcript (aMEF). Previously, Gundy et al.[7] showed that this asRNA is hardly or not detectable in *E. coli* wildtype strains, but that it becomes visible in a RNase III mutant strain. This indicates that the asRNA is rapidly degraded by RNases and has a very low stability. Nevertheless, mutations in the asRNA promoter proved that the transcription of this asRNA has a regulatory effect[7]. There are strong indications for termination by TI exactly at the

**Fig. 3 Rifampicin sensitive termination (RST): Example for transcription interference by an asRNA based on the co-transcriptional decay case.** $t_1$—the RNAPs from the sense and antisense strand collide. This leads to a transcription termination of one of the two RNAPs. RNAPs transcribing a non-translated RNA have a higher termination probability. Due to the termination, in the steady state, the RNAP concentration is higher 5' of the asRNA TSS and because the asymmetric termination probability of sense and antisense RNAPs we see a rapid, steplike decrease in the RNA concentration close to the asRNA TSS. $t_2 + t_3$—rifampicin stops the transcription initiation at the sense and antisense TSSs. Without the collision with the antisense RNAPs, all remaining sense RNAPs can now reach the positions after the asRNA TSS. This resembles an increase in positional synthesis rate and an increase of the RNA concentration after rifampicin addition for positions after the asRNA TSS. The delay of the increase depends on the time that the first RNAP of the "high-concentration wave" needs to reach the respective position. $t_4$—After the last RNAP has passed the respective position, the exponential decrease begins.

position of aMEF (Fig. 4a). Termination by TI does not require the accumulation of a regulatory RNA and the stability of the asRNA can be low. This limits the detectability of the antisense transcription but not its regulatory function. Furthermore, at collision, RNAPs transcribing the asRNA are more likely to terminate than the RNAPs transcribing the translated mRNA counterpart[10,11]. In transcription simulations of mRNA and asRNA pairs with an asymmetric termination probability, the asRNA concentration rapidly declines with the distance from its TSS (Fig. 5c, Supplementary Fig. 10). The RNA concentration reflecting the actual synthesis rate is only measurable in a short window <10 nt. In the example, the asRNA counts have dropped by >50% after 50 nt and to ~0 after 100 nt. In consequence, if RNA-seq workflows discriminate against short RNAs, asRNAs involved in TI by collision are likely underrepresented and the measured read counts do not reflect the actual synthesis rate.

**General detectability of RST events: Is this only the tip of the iceberg?** Several asRNAs were not associated with an RST signature. That might have biological explanations, e.g. that mRNA and asRNA are not expressed simultaneously in the same

individual cells, or that TI is prevented by other means. Moreover, the method can only detect a subset of RST instances. Assuming that an RST event is visible if it results in a post-rifampicin abundance-increase of at least 7.5%, this renders a wide range of RST events undetectable. Various parameters determine the shape of the abundance increase (Supplementary Fig. 11). The elongation rate and the position of the RST define the available time for the increase, the decay constant determines the time until steady state is reached and the termination rate sets the maximum possible abundance increase. Figure 5d illustrates the detectability of RST events for a high termination rate of 0.5 and different elongation rates and RNA stabilities. In an example (elongation rate: 25 nt/s, half-life: 5 min) the RST site needs to be at least 843 nt upstream of the TSS to yield the threshold increase. Supplementary Fig. 12 shows that a wide range of terminations by TI or sRNAs that occur closely after a TSS (Supplementary Fig. 6d) will not result in a detectable increase in post-rifampicin abundance.

Furthermore, it is essential that the abundance increase is captured within the sampling time points. An RST event in the *tolA* gene was detected in the *K. aerogenes* dataset[3] but likely

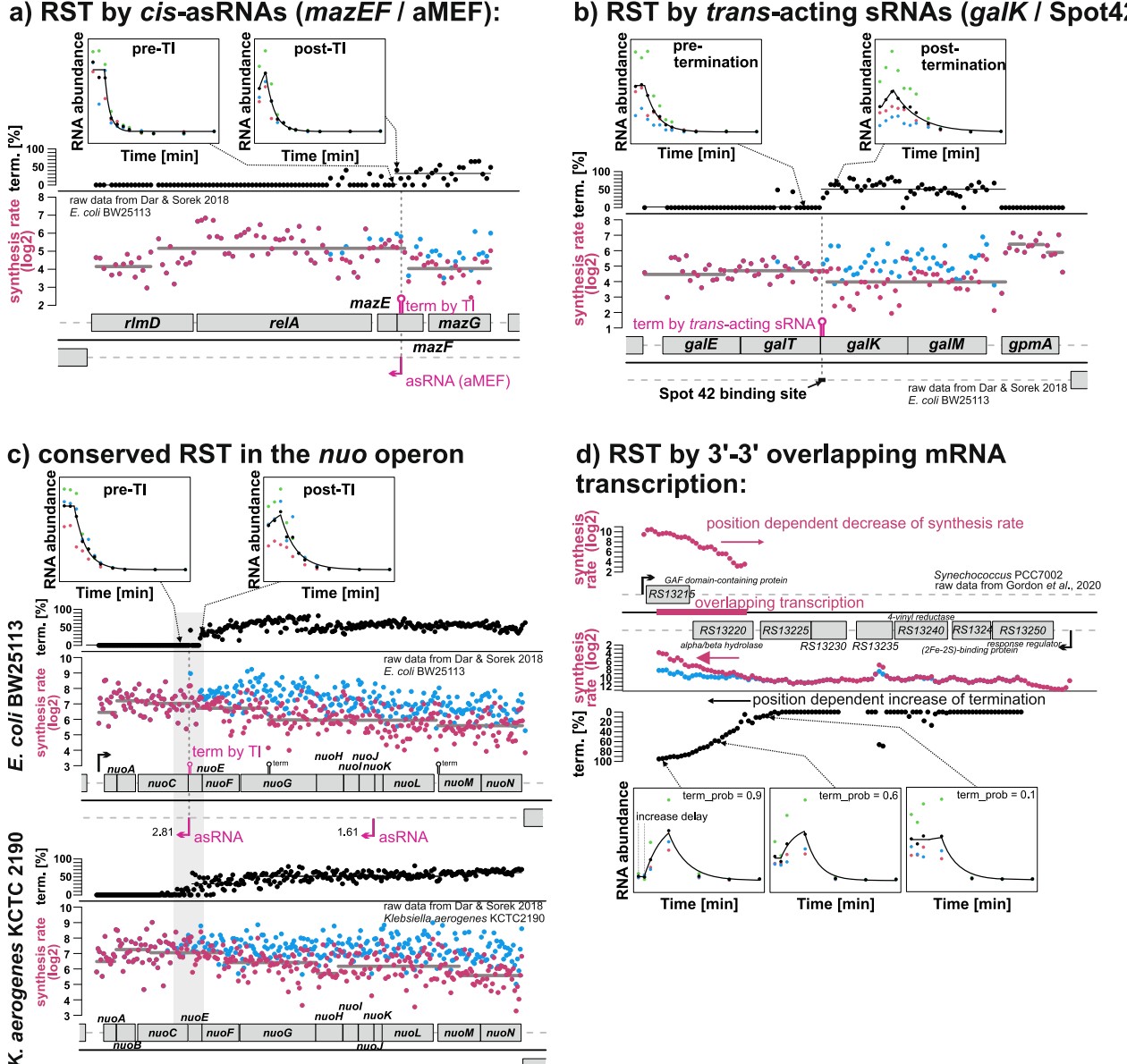

**Fig. 4 Examples of rifampicin sensitive termination (RST) caused by various means.** The log2 synthesis rate (dark pink dots) is displayed along the respective genomic segment. The synthesis rate is calculated from the RNA abundances at t = 0 and the fitted decay constants. The gray line indicates the mean log2 synthesis rate for a clustered segment. The blue dots show the calculated synthesis rate without RST. The black dots in the lane above or below show the estimated termination rate in %. Boxes beneath the plots: Fitted decay curves for selected pre- and post-RST bins and the respective raw data. The fit is based on three replicates (green, blue and pink dots), the mean of the replicates is indicated as black dots. **a** The pink arrow shows the asRNA TSS from Gundy et al.[7]. **b** Probable RST by the *trans*-acting sRNA Spot42 at the *galT/K* junction. **c** Probable RST by an asRNA in the *nuo* operon. The position of the RST is conserved between *E. coli* and *Klebsiella*. **d** Probable excludon in PCC 7002.

missed in *E. coli* MG1655 dataset[18] due to a lower time resolution (Fig. 5e, Supplementary Fig. 13). The temporal resolution is a critical aspect for RST detection. This is reflected in the lower number of RST instances for the datasets of *E. coli* MG1655 and *Synechocystis* (Fig. 5b), which both have a lower temporal resolution and lack e.g. a 1 min sampling point. We conclude that only a fraction of RST events are actually detected in this study and hence their frequency and biological relevance are likely underestimated.

**High-fidelity RNA stability calculations must consider the transcriptome architecture.** In the simplest case, a gene or operon is transcribed into a single transcript with constant properties along its entire length. In reality, iTSSs, partial termination events and processing or stabilization sites can lead to transcript species with different lengths and stabilities (Fig. 6). These different transcript species do not necessarily follow the respective gene annotations and an annotation-based analysis would not lead to exact stability estimates. The 'rifi' approach aims to infer the complex transcriptome architecture from the data. First, we fit the local features of 50 nt bins (RNA-Seq) or individual probes (microarray). The fits yield information about the delay, the decay constant (half-life) and, in case of RST, the termination rate, for each bin or probe. The RNA abundance before addition of rifampicin (t = 0) is another valuable piece of information. Based on these data we combine bins that belong to distinct transcripts with unchanged transcription parameters by a

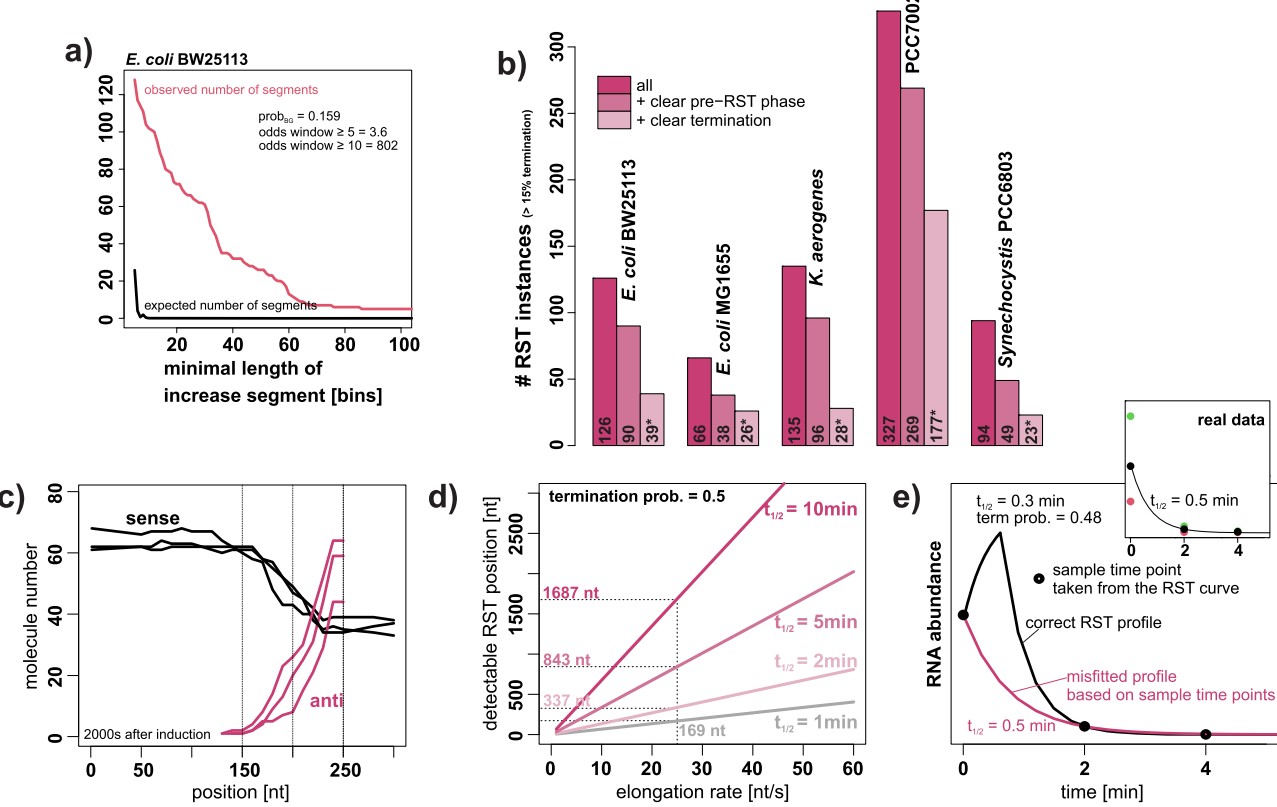

**Fig. 5 Evaluation of post-rifampicin abundance increases. a** Expected (in case of random noise) and observed numbers of segments with an post-rifampicin abundancy increase in the *E. coli* BW25113 dataset[3]. If post-rifampicin-increases are due to technical noise, they should be randomly distributed across all reads/probes. In contrast a systematic effect, such as TI, should affect consecutive reads/probes after an RST event (for details see methods). 7420/46803 bins from *E. coli* BW25113 are tagged as potential RST probes resulting in a background probability of $prop_{BG} = 0.159$. More details next to Supplementary Fig. 7. **b** Bar plot of the total number of RST instances for the respective datasets/organisms. "All": all RST segments with an average RST termination rate of ≥15%, a length of at least 5 bins and at least 75% RST bins, "+ clear pre-RST phase": Subset from all cases for which the segmentation identified a clear pre-phase without RST based termination. "+ clear termination": subset of the "+ clear pre-RST phase" instances, which are associated with an independently identified termination event. *numbers after manual subtraction of probable artifacts. **c** Simulation of positional abundances of sense mRNA and asRNA expression (synthesis rate$_{sense}$= synthesis rate$_{anti}$ = 0.6 molecules/s, elongation rate = 20 nt/s, λ = 0.01, term_prob$_{sense}$ = 0.3, three replicates). **d** Estimation of RST detectability dependent on elongation rate and RNA stability at a high term_prob$_{sense}$ of 0.5. The *y*-axis shows the minimal distance of the RST position from the TSS that ensures a post rifampicin abundance increase of at least 7.5%. The dotted lines indicate the lowest detectable distance from the TSS at an elongation rate of 25 nt/s for different RNA stabilities. **e** Example for RST that is not detectable, due to a relatively low time resolution. The example is based on a probable RST instance in the *E. coli tolA* gene. The black line shows the calculated solution for RST (term_prob = 0.48) in combination with a high elongation rate (60 nt/s) and a low stability (λ = 2.31 1/s, $t_{1/2}$ = 0.3 min). The pink curve shows the fit based on the available time points (0, 2, 4 min, taken from the calculated RST curve) that misses the RST due to low temporal resolution. More information in Supplementary Fig. 13.

dynamic programming algorithm. This segmentation is used by 'rifi' to identify transcriptional events such as iTSSs and (partial) termination sites between consecutive transcript segments and to define continuous transcriptional units (workflow summarized in Supplementary Fig. 14, 15). All full genome visualizations are available in Supplementary Data 3.

**Consequences for the inference of RNA stabilities and half-lives.** For *E. coli*, we found in general a good correlation between the 'rifi' results and the published results[3,18] (Supplementary Fig. 16a, b). However, there are ~90 transcripts with a wrong stability estimate due to the previous annotation-based workflow in *E. coli* MG1655 (Fig. 7a). The *miaA* gene is covered by 3 transcript segments of very different stability ($t_{1/2}$:0.83, 3.45 and 6.14 min, Fig. 7a, c). Other examples are *tgt* or the *ftsI* gene and its stable, gene-internal processing product, the FtsO sRNA sponge[35] (Fig. 7b, d). The separate half-lives of the *ftsI* mRNA and the FtsO sRNA were estimated to be 0.6 and 12.8 min, respectively, showing a vast difference in stability. Previous averaging over the *ftsI*

annotation led to a combined half-life of 8.4 min[18], which deviates by more than one order of magnitude from the actual half-life of the mRNA. We compared genes covered by two or more transcripts with different half-lives with the results from the Moffit et al. dataset[18] (Fig. 7a) to show that this situation leads to significant and frequent half-life deviations. The same was true for all other investigated organisms (Table 2, Supplementary Data 4). A good correlation was also seen for *Synechococcus* data[19] although the originally used model did not consider the delay leading to a slight overestimation of the half-life. This can be seen as a general trend in the scatterplots, where data points are slightly shifted towards higher half-lives for the original study (Supplementary Fig. 16c, d). This effect is more problematic for longer transcripts. Supplementary Figure 16g shows a ~8600 nt long TU with a delay of ~6 min for the last gene. For an example-bin within this gene, a model without delay misinterprets the delay as stability and calculates a half-life of 5.3 min versus the delay-adjusted half-life of 1.3 min. With 'rifi' we can compare the fragment-based half-life distributions of different organisms using a common standardized

**Table 1 Selected examples and possible reasons for instances of RST in *E. coli* BW25113 mentioned in the text.**

| 5′ gene | 3′ gene | possible reason | termination [%] | adj. *p*-value | Strand | Position BW25113 | RST event conserved in: |
|---|---|---|---|---|---|---|---|
| *galT* | *galK* | Spot42 | 56 | $1.15 \times 10^{-26}$ BW | − | 785426 | BW, KL$_w$ |
| *matP* 3′UTR | – | ompA 3′−3′ | 84 | $4.29 \times 10^{-12}$ BW | + | 1014500 | BW, MG$_w$, KL |
| *yebK* | *pykA* | DicF | 35 | $6.82 \times 10^{-9}$ BW | + | 1931975 | BW$_w$ |
| *nuoC* | *nuoE* | asRNA$_{T,D}$ | 38 | $1.15 \times 10^{-15}$ BW | − | 2395476 | BW, KL |
| *iscX* | *pepB* | asRNA$_{T,M}$ | 47 | $2.27 \times 10^{-10}$ BW | − | 2649876 | BW, MG |
| *mazE* | *mazF* | asRNA$_{T,G}$ | 34 | $2.13 \times 10^{-7}$ BW | − | 2904476 | BW, MG |
| *dam* | *rpe* | asRNA$_T$ | 47 | $1.66 \times 10^{-16}$ BW | − | 3508526 | BW, MG, KL |
| *malP* | *malQ* | asRNA$_{T,G}$ | 53 | $1.29 \times 10^{-22}$ BW | − | 3543526 | BW |
| *rbsC* | *rbsK* | RybB | 38 | $5.43 \times 10^{-8}$ BW | + | 3929575 | BW |
| *sdhB* | *sucA* | asRNA$_T$ | 46 | $2.93 \times 10^{-13}$ BW | + | 754075 | BW, MG$_w$ |
| *narG* | *narH* | | 56 | $1.02 \times 10^{-8}$ BW | + | 1277075 | BW, KL |
| *nirB* | *nirC* | | 82 | $1.43 \times 10^{-5}$ BW | + | 3489725 | BW, MG |
| *glgB* | *glgB* | | 63 | $1.08 \times 10^{-10}$ BW | − | 3566376 | BW, KL |
| *fruA* | *fruA* | | 57 | $3.42 \times 10^{-22}$ BW | − | 2254676 | BW, KL$_w$ |
| *rplL* | *rpoB* | | 48 | $5.85 \times 10^{-15}$ KL | + | – | KL, MG |
| *gsiA* | *gsiB* | | 38 | $1.51 \times 10^{-19}$ KL | + | – | KL, BW$_w$ |

The asRNA sites are from the Supplementary Table 3 in Thomason et al., 2014[37] (T), the *E. coli* BW25113 dataset[3] (D), Gundy et al. [7]. (G) or the MG1655 dataset[18] (M). *P*-values were calculated by t-test on the distributions of the RST-termination rates before and after the RST event and adjusted for multiple testing by the method of Benjamini Hochberg (BW: *p*-value from *E. coli* BW25113; KL: *p*-value from *K. aerogenes*). The final column shows if the RST event is conserved between the two *E. coli* strains and/or *K. aerogenes*. BW = *E. coli* BW25113, MG = *E. coli* MG1655, KL = *K. aerogenes* KCTC 2190. A subscript "w" in the conserved column indicates that an RST instance was detected by 'rifi' but did not match all required filter criteria.

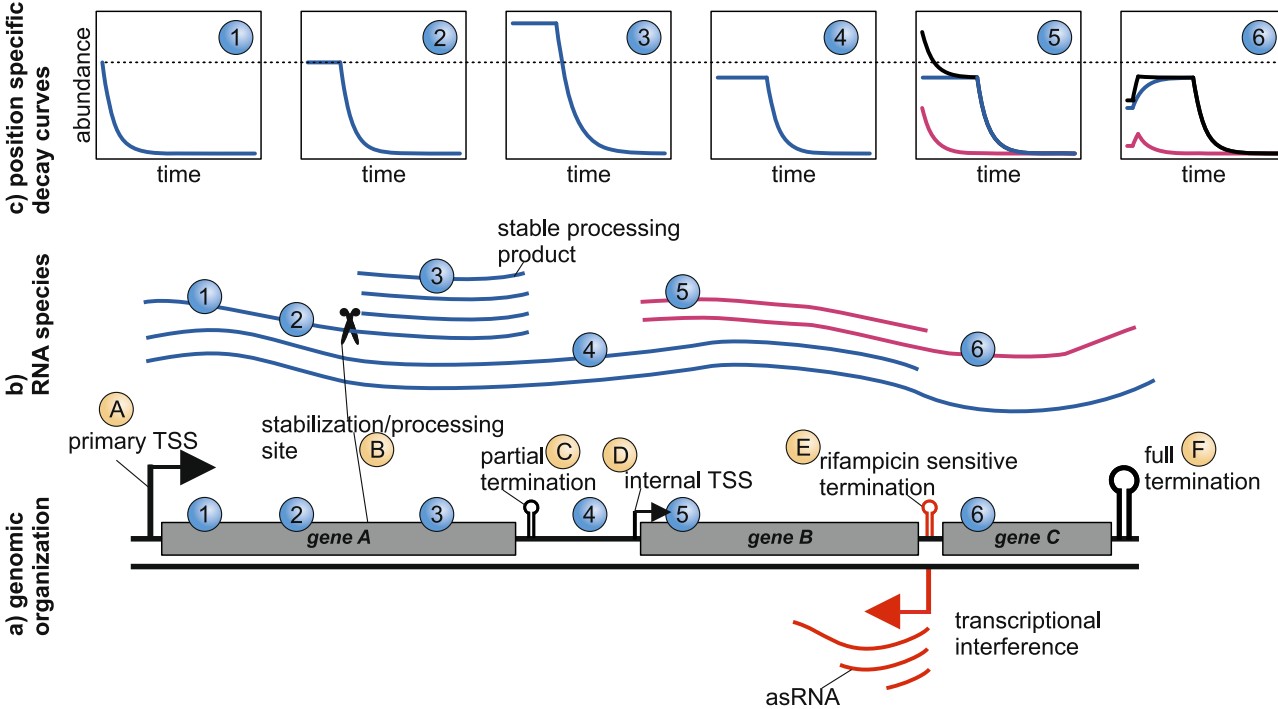

**Fig. 6 Addressing the complex transcriptome architecture.** Schematic example for a complex transcriptome architecture. **a** An operon consisting of three genes with its primary TSS (A), a partial termination site (C), an internal TSS (D), a *cis*-asRNA resulting in termination by TI (E) and the final full termination (F). The 3′ part of gene A is more stable than the 5′ part due to a processing or stabilization site (B). The transcriptional events A–F are automatically detected by 'rifi' as explained below and in Supplementary Fig. 14. **b** The architecture results in various overlapping transcripts. **c** The corresponding decay curves for different positions within the operon. (Position 1) The classic exponential decay at the start of the operon. (Position 2) A delayed exponential decay at a higher distance from the TSS. (Position 3) A delayed exponential decay with a shallower decline indicating a higher stability. Furthermore the RNA abundance at $t = 0$ is higher than in the previous curves (dotted line). In this example the abundance increase is explained by the stability increase and no iTSS is assumed. (Position 4) Delayed exponential decay with the same decline as at positions 1 and 2 but with a lower RNA abundance at $t = 0$. Here the lower RNA abundance is not explainable by changes in the stability, which indicates a partial termination. (Position 5) Only the combination (black curve) of the transcripts coming from the primary TSS (blue, long delay) and the transcripts coming from the iTSS (pink, short delay) is actually measureable. The increase in the abundance at $t = 0$ without a stability change and a potential reduction of the delay indicate an iTSS. (Position 6) The combined curve (black) for the transcripts coming from the primary and the internal TSS shows a post-rifampicin abundance increase, indicating TI. Also, the abundance at $t = 0$ is lower than for the positions located more 5′, at constant stability, this indicates termination.

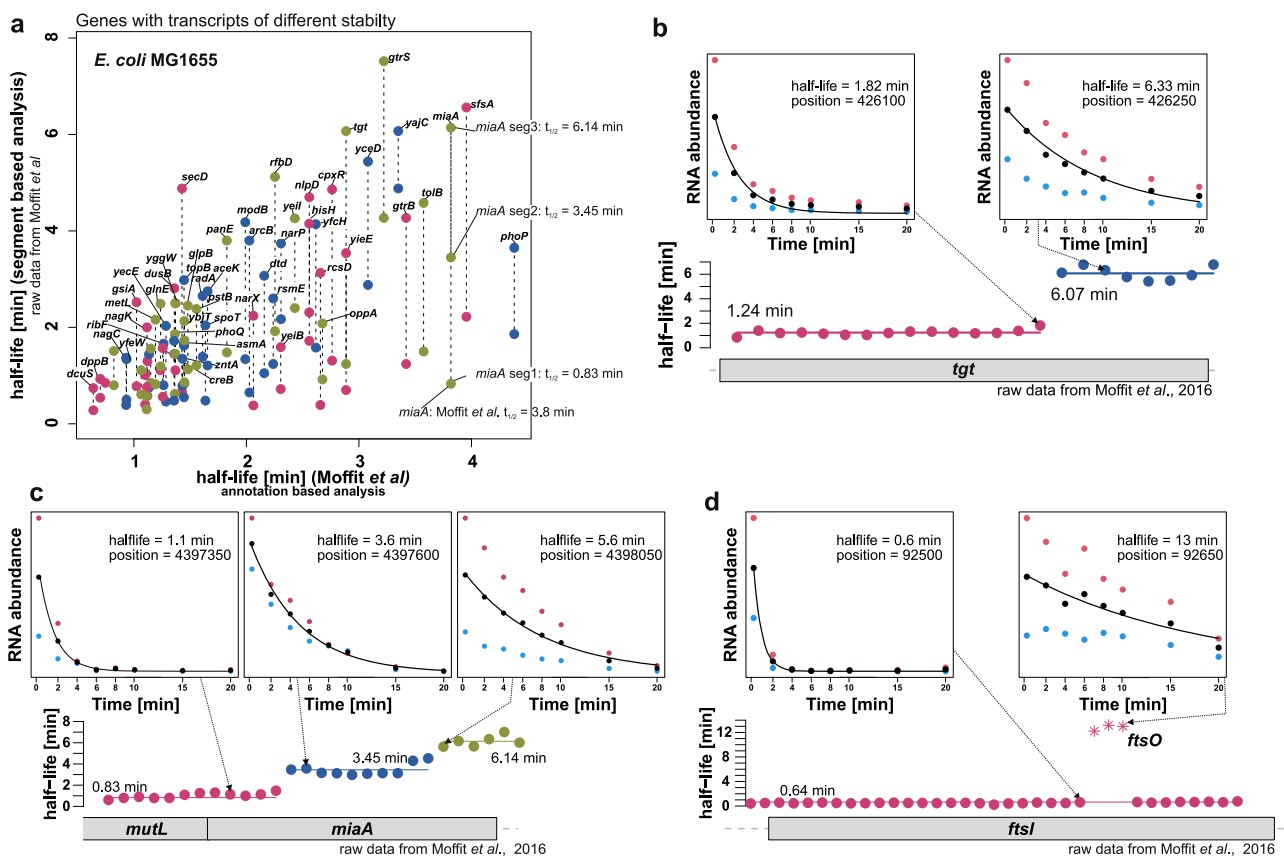

**Fig. 7 Improved half-live estimates by considering transcriptome complexity. a** Genes covered by two or more transcripts of different half-lives, for which the gene annotation-based half-life calculation leads to incorrect results. The plot compares the results from the original, annotation-based analysis, from Moffit et al.[18]. (*x*-axis) with the annotation independent workflow from this study based on the same raw data (two replicates). The following sub-panels show example genes. Each dot represents the fitted half-life in minutes for a 50 nt bin. The colors and the lines indicate the automatically segmented half-life fragments. The upper plots show the raw data and decay curves for a bin in the low stability fragment and a bin in the high stability fragment. The data from the replicates are colored (red, blue) and the mean abundance is given as black dots. The black line represents the fitted decay curve. **b** The *miaA* gene is covered by 3 transcripts with different stabilities. **c** The *tgt* gene is covered by 2 transcripts. **d** The stability for the *ftsI* gene and the stable internal FtsO sRNA is shown.

**Table 2 Overview about the findings for the different investigated datasets.**

| Organism | E. coli BW25113 | E. coli MG1655 | K. aerogenes KCTC 2190 | Synechococcus PCC 7002 | Synechocystis PCC 6803 |
|---|---|---|---|---|---|
| Platform | RNA-Seq | RNA-Seq | RNA-Seq | RNA-Seq | Microarray |
| Sampling time points [min] (after rifampicin addition) | 0,1,2,3,4,5,6,8,10,15 | 0,2,4,6,8,10,15 | 0,1,2,3,4,5,6,8,10,15 | 0,0.5,1,2.5,5,7.5,10 | 0,2,4,8,16,32,64 |
| Culturing parameters | LB-media, 37 °C, shaking, $OD_{600} = 0.5$ | LB-media, 32 °C, shaking, $OD_{600} = 0.4$ | LB-media, 37 °C, shaking, $OD_{600} = 0.5$ | A$^+$-media, 37 °C, 215 µE, air bubbling, $OD_{730} = 0.2$ | BG11-media, 30 °C, 50 µE, shaking, $OD_{750} = 0.7$ |
| Predominant mode of decay | | | co-transcriptional | | |
| High confidence RST events[a] | 39 | 26 | 28 | 177 | 23 |
| Global median half-life [min] | 1.53 | 1.65 | 1.01 | 0.78 | 0.93 |
| Global mean elongation rate [nt/s][b] | 33.2 | 27 | 28.5 | 42.1 | 22 |
| Genes covered by transcript segments of different stability[c] | 90 | 89 | 46 | 206 | 293 |
| Differential operon decay[d] | 131 | 138 | 62 | 82 | 71 |
| raw data from | Dar and Sorek[3] | Moffit et al.[18] | Dar and Sorek[3] | Gordon et al.[19] | This study |

[a]Manually curated.
[b]Excluding segments with a calculated elongation rate > 120 nt/s.
[c]Does not consider if the different segments belong to the same TU, i.e. the segments with different half-lives could be also shaped by terminators or iTSSs.
[d]Only sites with a log2 synthesis rate FC ≥ −0.5 ≤ 0.2 are considered in order to distinguish stability events from termination and iTSSs (*p*-value ≤ 0.05, |log2 half-life FC| ≥ 1 min). Culturing parameters: growth parameters and sampling optical density (OD). µE = µmol photons m$^{-2}$ s$^{-1}$. For more details regarding the experimental parameters please refer to the original publications.

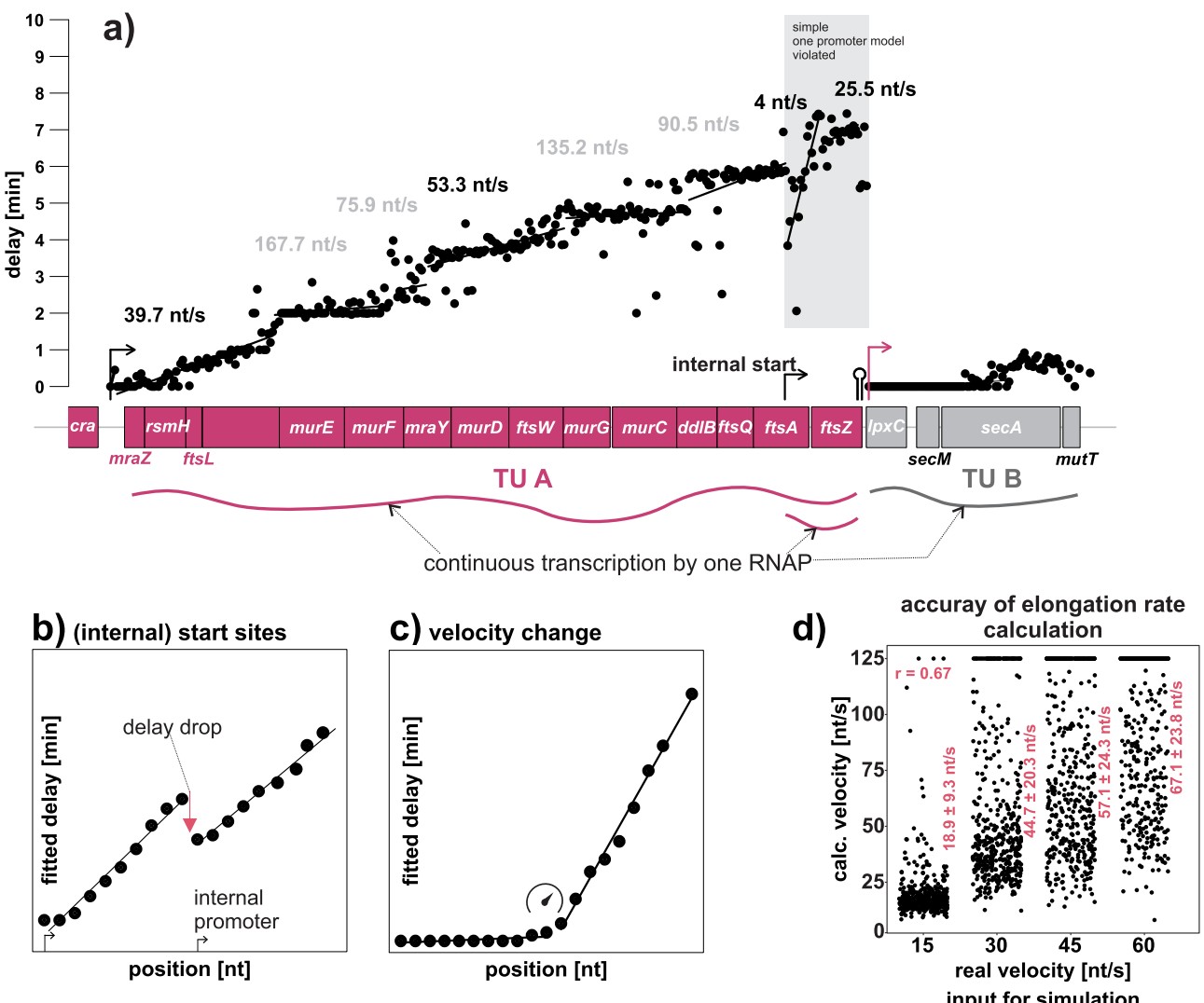

**Fig. 8 Classification of contiguous transcriptional units from the delay coefficient. a** Example for a long transcriptional unit in *E. coli*. The delay for the pink TU-A is steadily increasing from *mraZ* to *ftsZ*; this indicates the existence of a contiguous transcript. There is a drop in the delay in the middle of *fstA* indicating an iTSS. Transcription from both promoters overlap. At *lpxC* the delay starts again at 0 min, indicating that the previous transcript(s) were terminated and a new non-overlapping TU starts. Transcript regions with a continuous linear increase of the delay are clustered by 'rifi' and connected by a black trendline. The estimated transcription velocity is written above the respective fragments (un-physiologically high velocities in gray). In the shaded gray area the simple one promoter model is violated and the fitted delay is a combination from two individual transcripts. Using dynamic programming, 'rifi' automatically detects potential events such as TSSs (**b**) and velocity changes (**c**). **d** A simulation was done to assess the general accuracy of the elongation rate estimate. Data are based on the simulations with 36 parameter combinations (half-life: 1, 2 or 3.33 min; velocity: 15, 30, 45 or 60 nt/s; synthesis rate: 0.2, 0.4 or 0.8 molecules/s, 3 replications) and a continuous transcription without a parameter change. The jitter plots show the real input velocity used for the simulation versus the calculated velocity based on the fits. The Pearson correlation *r*-value and the mean fitted velocities with their standard deviation are given in the plots. The fragments with a biologically unrealistic velocity higher than 120 nt/s were excluded from the mean, sd and correlation calculation.

workflow. All investigated organisms have a median half-life below 2 min (Supplementary Fig. 16e, Table 2). Both *E. coli* datasets are similar with the highest median half-life of 1.53 and 1.65 min, while *Synechococcus* has the lowest median half-life of 0.78 min. The median half-life for *Synechocystis* was 0.93 min.

**Delay based estimates: Transcriptional units, internal start sites and transcription velocity.** The delay is dependent on the distance of a given position from the TSS and the elongation rate[22]. Assuming a constant elongation rate, the delay should linearly increase from the TSS to the end of the transcript (Fig. 8a, Supplementary Fig. 17a). Deviations from the linear

delay-increase indicate transcriptional events such as iTSSs (Fig. 8b) or velocity changes (Fig. 8c). The higher the differences in delay the higher the synthesis rate differences between consecutive bins. Thus, a steep drop suggests termination of the previous transcript or a considerably stronger new promoter. This connection makes the delay coefficient valuable for classifying contiguous transcriptional units (TUs), where a steep delay drop between two segments indicates the start of a new TU, while a minor drop implies an iTSS. It is important to consider that delay-based estimates rely on a low number of data points due to the relatively low temporal resolution of rifampicin data. This is evident in the tendency of the delay to be fitted at distinct points in the time series (Supplementary Fig. 18a). Chen et al.[22]. used the

delay to calculate the transcription elongation rate. We tried to estimate the expected accuracies for the elongation rates based on simulated data. The calculated velocities are clearly correlated with the actual input velocities but show a significant dispersion (Fig. 8d, Supplementary Methods). In conclusion, given the time resolution, the delay-based elongation rates for individual transcripts are rather rough estimates. A high number of the fitted transcript segments in all organisms have a calculated elongation rate >120 nt/s. We excluded these segments for a global comparison of the elongation rate distributions of the investigated organisms. The highest mean elongation rate was calculated for *Synechococcus* (42.1 nt/s) (Supplementary Fig. 18d). *Synechocystis* has the lowest mean elongation rate (22 nt/s) and *E. aerogenes* (28.5 nt/s), E. coli MG1655 (27 nt/s) and *E. coli* BW25113 (33.2 nt/s) lie in between. The mean elongation rate of 27 nt/s from *E. coli* MG1655 is close to the previously reported 25 nt/s[22].

Actually, there are many iTSS known within operons and genes[36–38]. This leads to overlapping transcription processes from transcripts initiated from different starting sites (Fig. 6), which violates the simple delayed-co-transcriptional decay model. Fitting those cases with the simple model can lead to an "artificial" delay (Supplementary Fig. 17b–d), which lies between the delay of the two independent transcription processes (*ftsA*, Fig. 8a). In theory, the model could be adapted to two or more overlapping transcripts. However, the fit with the wrong simple model and the solution for the two-promoter model are very close (Supplementary Fig. 17d). It needs to be considered that the "artificial" delay can also lead to an incorrect elongation rate calculation. An example is the calculated elongation rate of 4nt/s for the transcript segment after the iTSS in *ftsA* (Fig. 8a). The impact of internal TSSs and velocity changes on the fit of decay curves and the segmentation accuracy (Supplementary Fig. 17e–h and Supplementary Fig. 18b, c) are discussed in greater detail beside Supplementary Figs. 17, 18.

**Changes in the synthesis rate: Start and termination**. RNA abundance changes can result from alterations in stability and/or synthesis rate. Assuming RNA abundance reached a steady state prior to rifampicin addition, synthesis rates can be calculated from fitted decay constants. An increase in the synthesis rate ratio between a 3′ fragment and its 5′ segment (log$_2$ synthesis rate ratio: $R_{synt} > 0$) indicates the presence of a new iTSS (Fig. 9b, f). Conversely, a decrease ($R_{synt} < 0$) suggests (partial) termination (Fig. 9c, e) and an $R_{synt} \approx 0$ implies that observed changes are solely due to differences in transcript stability (Fig. 9a, d). Figure 9e illustrates a termination site at the 3′ end of *lpxC*, closely matching a site identified by term-seq[3]. The $R_{synt}$ of −3.47 corresponds to approximately 91% termination efficiency at this site. Similarly, a suggested TSS at the 5′ end of *secM* matches a start site from dRNA-Seq[37]. The 'rifi' synthesis-ratio-based terminator and TSS estimates demonstrated a significantly better-than-random overlap with results from term-seq[3], dRNAseq[3] and data from RegulonDB[39] (Supplementary Fig. 19, Supplementary Methods). Despite the relatively low resolution of the used 50 nt bins, 'rifi' also detected terminators in 5′ regions encoding small leader peptides (*thrL*, *pheL*, *mgtL*) or containing a riboswitch (*btuB*). Supplementary Figure 20 shows examples for 'rifi' terminator sites.

**Differential RNA decay within an operon**. Processing sites, stabilizing motifs, secondary structures or varying translation efficiencies can lead to fragments with different half-lives within the same operon[3]. This is likely an important aspect of bacterial gene regulation, but the identification of differential operon decay requires knowledge about the actual operons or more precisely about continuous TUs that start from the same TSS. For *E. coli*

such operon lists are available[3,39], but that might not be true for other organisms of interest. 'rifi' uses a bootstrapping approach to identify the TUs and the differential decay sites from the same data and calculates the stability differences between consecutive half-life segments within a TU. With this automatic approach we recovered 36/49 (Supplementary Data 4) previously described processing sites[3]. Two of the previous sites were not segmented with the used segmentation penalties, for 8 sites the expression was low and the reads were filtered prior to the analysis and for three sites we detected similar half-life differences but the segments were predicted to be in different TUs. From the latter three cases, two TU borders are likely artifacts from the delay calculation, but one TU border between *gpmM* and *envC* is strongly supported by dRNA-Seq data[35] (Supplementary Fig. 21). Beyond the previously reported cases 'rifi' detects 95 additional cases of differential operon decay in *E. coli* and many cases in the other investigated organisms (Table 2, Supplementary Data 4; examples: *cysW/A*, *minC/D*, *napD/A*, Supplementary Fig. 21).

**Conclusion**

Extending on previous studies[22], we show that most, if not all, transcripts from *E. coli*, *K. aerogenes*, *Synechococcus* PCC 7002 and *Synechocystis* PCC 6803 can undergo degradation before the RNAP terminates (co-transcriptional decay). The decay mode can be determined directly from steady-state RNA abundance patterns in standard transcriptomic data, without requiring rifampicin time series. A constant RNA abundance throughout the transcript indicates co-transcriptional decay, while post-transcriptional decay is indicated by a linear decrease towards the 3′ end. Stochastic simulation reveals that the co-transcriptional decay patterns can result from 5′ exoribonucleases or a combination of endo- and 3′ exoribonucleases. Notably, *Synechocystis* and *Synechococcus* possess both 5′ exo and endoribonucleases[19,25,26], whereas *E. coli* is thought to primarily rely on endo- and 3′ exoribonucleases[27]. Co-transcriptional decay is consistent with coupled transcription-translation, which is common in many prokaryotes[40]. However, uncoupled transcription-translation is also frequent. In *B. subtilis*, transcription elongation outpaces translation, leading to uncoupling[41], and there are instances of spatiotemporal separation of transcription and translation[40]. Nevertheless, the contradiction of co-transcriptional decay and uncoupled transcription-translation is not as substantial as anticipated. As previously mentioned[22], co-transcriptional decay is a stochastic process and depending on the decay constant, transcript length and elongation rate, a given percentage of full-length transcript can be expected in the steady state. For a transcript length of 1000 nt (half-life: 2 min, elongation rate: 25 nt/s) approximately 79% of transcripts are not degraded before termination (Supplementary Fig. 22). 'rifi' is a versatile R-package applicable to high-resolution rifampicin time series datasets from various platforms (microarrays, RNA-Seq) and diverse organisms (Table 2). Using the *E. coli* MG1655 data[18], we illustrate hat considering the complex transcriptome structure leads to corrected half-life estimates for around 90 genes. This observation holds true for other organisms (Table 2). "rifi" automatically detects processing or stabilization sites that lead to differential stabilities within a transcriptional unit. It also detects transcriptional start- and termination sites based on calculated synthesis rate changes. This complements RNA-Seq-based approaches like dRNA-Seq[38] and term-seq[42] for defining transcriptional start sites (TSS) and terminators. Although the bin-based 'rifi' method lacks the resolution of RNA-Seq, it can integrate multiple data types to provide a more comprehensive understanding compared to sequencing-based methods. By analyzing synthesis rate changes 'rifi' distinguishes 3′ processing sites from termination sites and calculates actual termination rates.

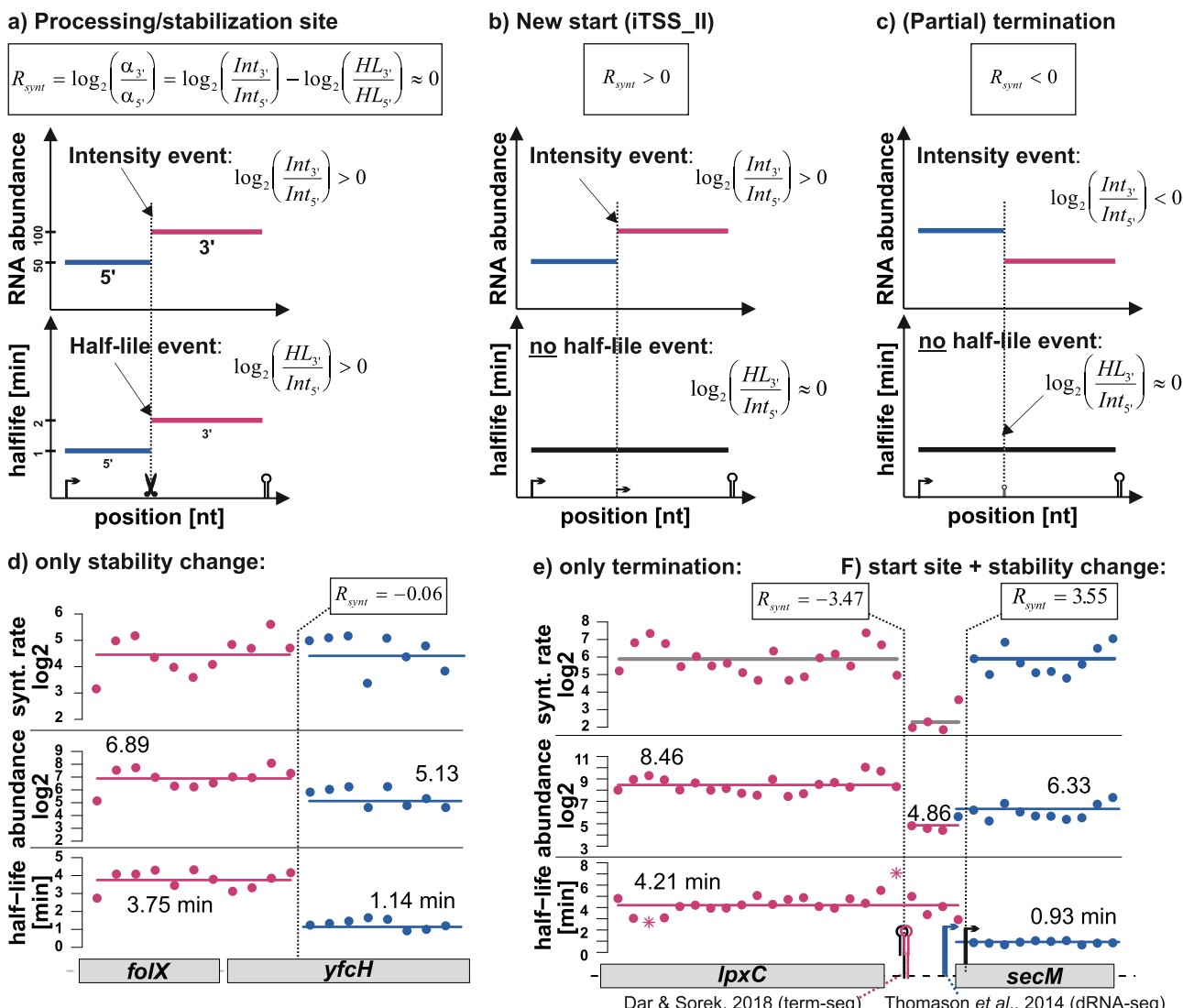

**Fig. 9 Detection of termination sites, TSSs and processing/stabilization sites.** Example scenarios are shown. For each pair of consecutive abundance fragments in an estimated TU the synthesis ratio $R_{synt}$ is calculated ($\alpha$ = synthesis rate [molecules/min], $Int$ = RNA abundance at $t = 0$ [counts], $HL$ = half-life [min], the indices indicate values for the 3′ and 5′ segment). **a** If $R_{synt} \approx 0$ the abundance differences are only due to changes in the transcript stability. **b** An $R_{synt} > 0$ indicates a new TSS. **c** An $R_{synt} < 0$ indicates a terminator. In panels B and C only examples with no change in transcript stabilities are shown. In reality mostly both, stabilities and synthesis rates, differ. The following data are based on the Dar and Sorek[3] dataset (3 replicates). **d** Estimated half-lifes, RNA abundances and calculated synthesis rates are plotted for each 50 nt bin. Colors and the mean-line indicate fragments from the dynamic programming (DP). Bins that are taken as outliers by the DP algorithm are drawn as stars. Example for two RNA fragments in the *folX-yfcH* operon that have different stabilities. The differences in the intensities are fully explainable by the different half-lifes. Note that the exact fragment borders are only detectable if the fragmentation appears independent of existing annotations. **e** Example for a termination site 3′ of *lpxC* which closely co-locates with a site estimated by term-seq[3] and **f** a TSS at the 5′ end of *secM* which co-locates with a start site estimated by dRNA-seq[37]. Note that the resolution of the rifampicin approach is rather low, as we use bins of 50 nt for the site estimation.

The delay of exponential decay serves as a measure to track uninterrupted transcription processes from the transcription start site (TSS) to the final termination site. The delay-based transcription unit (TU) definition remains unaffected by partial termination sites or internal processing sites and has the potential to detect iTSSs associated with parallel transcription processes. However, compared to the more accurate stability estimates, all delay-based estimates suffer due to the limited number of data points available for fitting the delay. A higher time resolution in sampling points following rifampicin addition would significantly improve the performance of velocity estimation, TU definition, and iTSS detection.

A surprising discovery of this study is that rifampicin time-series data offer a live view of TI by the collision of RNAPs and transcription termination by *trans*-acting sRNAs. These phenomena are indicated through decay curves that exhibit increased RNA abundance after rifampicin addition.

- Stochastic simulations confirm that these distinctive patterns are fully explainable by the biological phenomenon of RST, i.e. through TI or sRNA interaction, or pre-steady-state expression.
- Technical noise and a systematic normalization bias have been excluded as explanations for these patterns.
- RST patterns are observed in all five datasets representing four organisms from divergent phylogenetic groups (Cyanobacteria and Proteobacteria) (Fig. 5b, Table 2), independent of different experimental workflows.

– In *E. coli*, two instances of RST are supported by prior experimental findings. These include termination at the *mazE/mazF* junction likely due to TI by a cis-antisense RNA[7] and termination at the *galT/K* junction by the *trans*-acting sRNA Spot42[30].

– 14 potential RST instances are conserved between different *E. coli* strains and/or *Klebsiella*. This makes technical artifacts very unlikely, as the data are not only from different organisms but also from different research groups and partially based on different growing conditions.

The identified potential cases of RST likely represent only a fraction of the overall occurrences. Our model demonstrates that a considerable number of TI events would not lead to a noticeable increase in RNA abundance after rifampicin treatment if the termination site is located too close to the TSS. We present evidence supporting the significant impact of asRNAs on the differential expression of operons, despite their limited detectability. This finding contributes to the growing body of evidence suggesting that many asRNAs are not merely products of futile pervasive transcription but actively participate in gene regulation.

## Methods

**Models**. First, the decay curves for all individual bins/probes are normalized to the highest RNA abundance value in the time course and fitted with the appropriate model to extract the delay and the decay constant $\lambda$. The decay is then transformed into the half-life ($t_{1/2} = \frac{\ln(2)}{\lambda}$). In case of RST also the termination probability is extracted. The co-transcriptional decay is fitted by the previously described model[22] where the concentration of the RNA $c$ is dependent on the position downstream of the TSS $n$ and the time after rifampicin addition $t$, where $\alpha$ is the synthesis rate, $\lambda$ the decay constant and $v$ the elongation velocity. As we do not now $n$, we internally fit the delay ($delay = \frac{n}{v}$) directly. A graphical explanation of the models is given in Supplementary Fig. 23. Used parameters: $\lambda$ = decay constant [1/s], $\alpha$ = synthesis rate [mol/s], $v$ = elongation rate [nt/s], $n$ = position relativ to the TSS [nt], $t$ = time after rifampicin addition [s], $L$ = total transcript length [nt], $n_{term}$ = position of rifampicin sensitive termination [nt], $\beta$ = termination proportion:

$$c(t,n) = \begin{cases} \frac{\alpha}{\lambda} & \text{if } t < \frac{n}{v} \\ \frac{\alpha}{\lambda} \times e^{-\lambda(t-\frac{n}{v})} & \text{if } t \geq \frac{n}{v} \end{cases} \quad (1)$$

For microarrays, we mostly use a variant of the co-transcriptional decay model that accounts for a constant background ($bg$). The non-BG model is only used for probes where the RNA abundance at the last available time point is above a definable background threshold:

$$c(t,n) = \begin{cases} bg + \frac{\alpha}{\lambda} & \text{if } t < \frac{n}{v} \\ bg + \frac{\alpha}{\lambda} \times e^{-\lambda(t-\frac{n}{v})} & \text{if } t \geq \frac{n}{v} \end{cases} \quad (2)$$

Depending on the decay pattern and information about the data generation, i.e. microarray or RNA-Seq, different fitting models are used. In regions of RST we use a dedicated model. We furthermore distinguish between a model without background (BG) as it is suitable for RNA-Seq and a model with BG for most probes from microarrays. Fitting with or without BG can make a considerable difference, especially when a stable transcript has not yet reached the baseline at the last measured point in time. In this case, a model with BG will misinterpret the RNA abundance at the final time as BG and the estimated half-life will be too low. An extreme case is *grxB* from the *E. coli* MG1655 dataset (Supplementary Fig. 16f); the original fit was done with a BG model resulting in a half-life of 0.87 min[18], while the fit without BG estimates 26 min, a roughly 30 fold deviation. Vice versa, a fit without BG would result in artificially high stabilities in data which do have a background, as e.g. data from microarrays.

In case of the post-transcriptional decay also the total length of the transcript $L$ is relevant. The concentration at $t = 0$ is the steady-state concentration plus the number of transcripts that are currently actively transcribed and contain the respective position. The below model corrects the model presented by Chen et al.[22], which models a constant position dependent RNA concentration in steady state before rifampicin addition instead of the actual position dependent abundance decline (Supplementary Fig. 2):

$$c(t,n,L) = \begin{cases} \frac{\alpha}{\lambda} + \alpha \frac{L-n}{v} & \text{if } t < \frac{n}{v} \\ \frac{\alpha}{\lambda} + \alpha \left( \frac{L}{v} - t \right) & \text{if } \frac{n}{v} \leq t < \frac{L}{v} \\ \frac{\alpha}{\lambda} + \times e^{-\lambda(t-\frac{L}{v})} & \text{if } t \geq \frac{L}{v} \end{cases} \quad (3)$$

If rifampicin is added in the pre-steady-state time window, meaning that the time of first transcription initiation $t_{ini}$ is smaller than the time of rifampicin addition $t_{rif}$ and that the time between the start of the transcription and the

rifampicin addition $t_{rif}$ is smaller than the time required to reach the steady-state concentration $t_{steady}$, the concentration is ($t$ is always relative to $t_{ini}$):

$$(t,n) = \begin{cases} 0 & \text{if } t < \frac{n}{v} \\ \frac{\alpha}{\lambda} - \frac{\alpha}{\lambda} \times e^{-\lambda(t-\frac{n}{v})} & \text{if } \frac{n}{v} \leq t < \frac{n}{v} + t_{rif} \\ \left( \frac{\alpha}{\lambda} - \frac{\alpha}{\lambda} \times e^{-\lambda(t-\frac{n}{v})} \right) \times e^{-\lambda(t-\frac{n}{v})} & \text{if } t \geq \frac{n}{v} + t_{rif} \end{cases} \quad (4)$$

In case of the rifampicin sensitive termination (RST), the concentration is dependent on the termination proportion $\beta$ and the position of the rifampicin sensitive termination $n_{term}$. Here, $\frac{n-n_{term}}{v}$ is the time that the polymerases need from the (relieved) termination site to position $n$, i.e. the delay of the increase. The maximal time the RNA abundance can increase is $\frac{n_{term}}{v}$, which resembles the maximal time window of the higher RNAP concentration for post termination site positions:

$$c(t,n) = \begin{cases} \frac{\alpha - \alpha \times \beta}{\lambda} & \text{if } t < \frac{n-n_{term}}{v} \\ \frac{\alpha}{\lambda} - \frac{\alpha \times \beta}{\lambda} \times e^{-\lambda\left(t-\frac{n-n_{term}}{v}\right)} & \text{if } \frac{n-n_{term}}{v} \leq t < \frac{n}{v} \\ \left( \frac{\alpha}{\lambda} - \frac{\alpha \times \beta}{\lambda} \times e^{-\lambda\left(t-\frac{n_{term}}{v}\right)} \right) \times e^{-\lambda(t-\frac{n}{v})} & \text{if } t \geq \frac{n}{v} \end{cases} \quad (5)$$

It is a frequent case that a position is passed by polymerases which have been started at two or more different TSSs. Here the measured RNA concentration/count/intensity is based on a mixture of transcripts of different lengths. Assuming a case with two promoters, were the more upstream promoter has the synthesis rate $\alpha_1$ and the downstream promoter $\alpha_2$, we can model the transcript number with the following equations. For simplicity we assume similar elongation velocities $v_1 \approx v_2 \approx v$ and degradation constants $\lambda_1 \approx \lambda_2 \approx \lambda$ for both types of transcripts. The distance to the first promoter is given by $n_1$ and the distance to the second promoter by $n_2$:

$$n_2 < n_1$$

$$c(t,n) = \begin{cases} \frac{\alpha_1}{\lambda} + \frac{\alpha_2}{\lambda} & \text{if } t < \frac{n_2}{v} \\ \frac{\alpha_1}{\lambda} + \frac{\alpha_2}{\lambda} \times e^{-\lambda\left(t-\frac{n_2}{v}\right)} & \text{if } \frac{n_2}{v} \leq t < \frac{n_1}{v} \\ \frac{\alpha_1}{\lambda} \times e^{-\lambda\left(t-\frac{n_1}{v}\right)} + \frac{\alpha_2}{\lambda} \times e^{-\lambda\left(t-\frac{n_2}{v}\right)} & \text{if } t \geq \frac{n_1}{v} \end{cases} \quad (6)$$

The lag of the action of rifampicin after addition to the media is between 5 and 10 s in *E. coli*.[21] Assuming that all RNAPs are equally affected by the lag, this should result in an ubiquitous delay of some seconds in the 5′ ends of all transcripts. For hypothetical data with a much higher temporal resolution in the time frame <30 s this ubiquitous delay would be detectable by the delayed co-transcriptional model. In reality the temporal resolution is much lower and an experimental error of some seconds in the time taking for the sample time points can be assumed. In conclusion, the small lag of the rifampicin action should not seriously affect the results.

**'rifi' workflow**. First, all regions with continuous coverage, i.e. regions with no coverage gap >300nt, are taken as initial segments. The further segmentation is done in a hierarchical manner. Starting on the delay, then on the half-life and finally on the RNA abundance before rifampicin addition, i.e. the assumed steady-state expression level. The operon structure and transcriptional events (e.g. start sites or termination sites) are estimated based on this initial fragmentation as described in the following (Supplementary Fig. 14). The general structure of the 'rifi' tool is visualized in Supplementary Fig. 15. The segmentation problem boils down to setting the right splits within the data[43]. Therefore, 'rifi' tries to minimize the residuals of the data points to a linear regression (delay) or the average (half-life, RNA abundance) of the whole segment. In order to prevent excessive segmentation, each split adds a penalty $P$ to the global segmentation score. Thus, the number of clustered fragments strongly depends on the penalties. Lower penalties lead to a stronger fragmentation and a higher sensitivity for event detection at the cost of a lower specificity. 'rifi' tries to automatically estimate the best penalties based on a sample of the given dataset. Different scoring functions are used to assess the similarity as described below. The optimal segmentation $S_{n,k}$ that splits a sequence of $n$ ordered elements in $k$ non-empty segments is defined by the recursion

$$S_{n,k} = \min_{n-1 \geq k \geq 1} \min_{i \leq n} S_{i-1,k-1} + f(i,n) + P \quad (7)$$

where $f(i,n)$ defines the optimal score of the segment from element $i$ to $n$. Mostly we use the variant where the total score $S_{n,k}$ is minimized but for the TU definition the score is maximized and the penalty $P$ is negativ:

$$S_{n,k} = \max_{n-1 \geq k \geq 1} \max_{i \leq n} S_{i-1,k-1} + f(i,n) - P \quad (8)$$

'rifi' consists of seven top level steps for data processing, at the core of which, the fitting of the input data and the fragmentation of the resulting data are conducted.

**'rifi_preprocess'**. 'rifi_preprocess' filters the data, assigns the bins to the correct fitting model (i.e. co-transcriptional model with or without background (BG) or the RST model) and performs the general processing of the data structure. For the

RNAseq data, bins with an average count below 10 were discarded and no further filtration was conducted. For the microarray dataset, probes were tagged for the BG model if the latest time point was below our BG threshold of 4000 A.U. Probes with the highest expression at the last time point or the lowest expression at the first time point were filtered. As the dynamic programming approach has a polynomial time complexity, it is paramount to divide the whole genome into smaller pieces by regions without expression. We segmented the genome at regions without expression over 300 nts (changeable parameter).

A two-step process assigns the bins to the RST fitting model. Bins influenced by RST are likely to show an increase in relative RNA abundance. Bins or probes for which at least one time point $t > 0$ (averaged over all replicates) shows a 7.5% higher abundance than time point $t = 0$, are labeled with a value $y$ of 1, while all other bins are labeled with $y = -1$. Dynamic programming is used to identify regions with predominantly probes that behave like they are under the influence of RST, to account for noise and irregularities. For this, a simple scoring function is used that calculates the sum of the absolute deviation from the mean of each fragment:

$$y \in \{-1, 1\}$$

$$f(i, n) = \sum_{i=1}^{n} \left| y_i - \frac{1}{n} \sum_{i=1}^{n} y_i \right| \tag{9}$$

A split penalty $P$ of 10 is applied in this segmentation. Segments with at least 75 % of such RST indicator bins are flagged as potential RST candidates and fitted with the RST model. Additionally, 1000 nucleotides before the potential RST event are likewise subjugated to the RST fit, to create a region of pre-RST. The fits are done using the R nls or nls2 functions with the appropriate models. The RST model introduces, compared to the simple co-transcriptional mode, 2 additional free parameters. For a termination proportion $\beta = 0$, the model boils down to the co-transcriptional model. The additional freedom in the RST model can favor a $\beta > 0$, also in the non-RST case, especially if the number of datapoints are low. On the other hand, a comparison of the RST with the non RST model by e.g. the Bayesian information criterion[40] would be too conservative; again because usually only few data points can be used to fit the increase. Here we use the workaround that we collect all fits in a given residual range compared to the best fit (default: 20%). From these fits we choose the fit with the smallest termination probability. Reducing the range increases the sensitivity to detect regions with RST. All analyses were done with rifi R-package version 1.2.2 from Bioconductor version 3.16.

**'rifi_penalties'.** Penalties for delay, half-life, RNA abundance, and TI fragmentation are highly dependent on the number of datapoints per nucleotide and thus the bin size for RNA-seq data or probe size and probe distribution for microarray data. 'rifi' tries to automatically estimate the best penalties based on a sample of the coverage segments in the given dataset. For this sample a range of penalties is tried, then for all consecutive fragments, 'rifi' tests if they are statistically significantly different. Finally, the penalty with the highest difference between significant- and non-significant splits is selected. ANCOVA and t-test are used to evaluate the splits for delay and half-life, abundance, and TI respectively.

**'rifi_fragmentation'.** The values extracted from the fitting (delay, half-life, and TI-termination factor) alongside the RNA abundance are all representative of a certain cellular behavior at the given position. By the various fragmentation steps, the bins are clustered into groups of similar behavior.

The bins are fragmented by the described dynamic programming in a hierarchical manner (Supplementary Fig. 14). Each position segment is fragmented into fragments of uniformly increasing delay, which in turn are then fragmented into fragments of similar half-life, which are finally fragmented into fragments of similar RNA abundance. From the delay fragments TUs are formed by the overall increase of the delay, on which TI fragments are formed.

**Delay segment scoring function.** To fit the increase in delay, a scoring function is used that calculates the sum of the absolutes of the residuals of the linear fit through all delay values $y$ over the positions of the bins $x$. The slope $u$ of the linear fit is restricted within biological meaningful borders between zero and 1/60. Individual noisy data points or small transcripts (covering less than the required 3 data points for being a separate segment) might interfere with the proper segmentation or force artificial splits. For that reason we use a scoring function that allows the exclusion of data points (outliers) from a segment. A maximal allowed number $O$ of outliers can be defined, by default maximal 0.4*n but not more than 10 outliers are allowed. If a segment with n data points is scored, the number of values $\{1, …, n\}$ considered in the fit will be reduced $O$ times in a manner where the value corresponding to the highest residual is excluded each round, resulting in the vector $o$ representing the indices of the elements that have the respective highest residuals.

Outlier selection:

$$o_j = \underset{i}{argmax} \left| y_i - u * x_i + t \right|; i \in (1, …, n) / \left\{ o_1, …, o_{j-1} \right\}; j > 0 \tag{10}$$

For the addition of each outlier, an outlier-penalty $N$ will be added to the score. The scoring function chooses the number of outliers $j = \{0, …, O\}$ that lead to the

minimal combined score. The minimal allowed length $m$ for a delay fragment is three.

Scoring function:

$$0 \le u \le \frac{1}{60}$$

$$f(i, n) = \underset{j}{min} \begin{cases} \sum_{i=1}^{n} |y_i - u * x_i + t| ; i \in (1, …, n) ; j = 0 \\ \sum_{i=1}^{n} |y_i - u * x_i + t| + P * j ; i \in (1, …, n) / \{o_1, …, o_j\} ; j > 0 \end{cases}$$

$$f(i, n) = \underset{j}{min} \begin{cases} \sum_{i=1}^{n} |y_i = u * x_i + t| i \in (1, …, n) ; j = 0 \\ \sum_{i=1}^{n} |y_i = u * x_i + t| + P * j ; i \in (1, …, n) / \{o_1, …, o_j\} ; j > 0 \end{cases} \tag{11}$$

**Half-life, RNA abundance and transcriptional interference.** The fragmentations of the half-life, RNA abundance and TI fragments all use the same scoring function that, similarly to the scoring function for the assignment for the TI fit, calculates the sum of the deviations to the mean. In addition, outliers are penalized here in the same way as described for the delay fit, resulting in the following scoring and outlier detection functions. The abundance values are calculated as their log2 to normalize for higher spread in the data at higher intensities. The minimal allowed length $m$ for a delay fragment is three:

$$o_j = \underset{i}{argmax} \left| y_i - \frac{1}{n} \sum_{i=1}^{n} y_i \right|; i \in (1, …, n) / \{o_1, …, o_{j-1}\}; j > 0 \tag{12}$$

$$f(i, n) = \underset{j}{min} \begin{cases} \sum_{i=1}^{n} \left| y_i - \frac{1}{n} \sum_{i=1}^{n} y_i \right|; i \in (1, …, n); j = 0 \\ \sum_{i=1}^{n} \left| y_i - \frac{1}{n} \sum_{i=1}^{n} y_i \right| + P * j; i \in (1, …, n) / \{o_1, …, o_j\}; j > 0 \end{cases} \tag{13}$$

**Transcription unit.** All delay fragments are combined into TUs by the distance between the delay at their last position and the delay of the first position of the consecutive fragment. A steep drop of the delay is more likely the initiation of a new TU, while a small drop implies an internal start (iTSS_I). If the delay does not revert back to ~0 min at the start of a new segment, in general overlapping transcription can be assumed. A delay fragment starting at zero minutes always initiates a new TU. In the current version 'rifi' defines TUs only based on the delay, which can be problematic if the delay difference between the final and the first segments of two TUs is too small for segmentation. This can happen for short TUs, or if the elongation rate is very high. In contrast to the other segmentations, here the score is maximized. The default split penalty $P$ is $-0.75$.

The scoring function sums up the distance between the delay at the ending points e of each fragment except the last one to the delay at the starting point s of each corresponding consecutive fragment in potential TU with $n + 1$ segments and $n$ splits. We furthermore assume that fragment-starts with a low delay, below 0.6 min, are more likely to initiate a new TU than fragment-starts with higher delays. For that reason, each start gets a delay-based modifier based on a logarithmic function. The formula causes delay values at the start of the second fragment to be penalized exponentially harsher the closer the value is to 0, while values higher than 0.6 are rewarded but only lightly. The final scoring function is

$$f(i, n) = \sum_{i=1}^{n} s_i - e_i + \sum_{i=1}^{n} \ln(s_i) + 0.5 \tag{14}$$

**Statistics and Reproducibility—'rifi_stats'.** The statistics for decay constant, delay and RST termination rate from the nonlinear-least-square model fits to the individual bins/probes are not used for a probe filtering or as weights for the dynamic programming. Fits were done on all available replicates simultaneously (*Escherichia coli* K12 BW25113[3]: 3 replicates, *Klebsiella aerogenes* KCTC 2190[3]: 1 dataset no replicate, *Synechococcus* sp. strain PCC 7002:[19] 3 replicates, *E. coli* K12 MG1655[18]: 2 replicates, *Synechocystis* PCC 6803: 3 replicates). All statistics are based on the distributions from the fitted values (delay, HL log2(abundance)) from individual segments. For the HL segments and the abundance segments the HL and abundance standard deviation and standard errors are calculated based on the distributions from the individual probes/bins belonging to the segments. The values are reported in the final 'rifi' results R-object. Consecutive HL and abundance segments are tested using a two-sided Student's $t$ test for a significant fold change. The log₂ fold-changes and $p$-values are reported in the 'rifi' output as "Int_event" and "HL_event". The log₂ synthesis rate ratio $R_{synt}$ is calculated between all consecutive abundance fragments within the same TU, with the synthesis rates $\alpha$, the RNA abundance before rifampicin addition $int$, the degradation constant $\lambda$ and the half-life $HL$. Due to the hierarchical segmentation strategy an abundance segment will never overlap two half-life segments, but the matching half-life segment might be bigger than it. In those cases, the half-life

**Table 3 Statistical tests used for the transcriptional events.**

| Type of fragment | Event | Test | Criterion |
|---|---|---|---|
| Delay | iTTS I | *t* test (two-sided, unpaired, unequal variance) | Difference between projected end of the first segment and start point of the consecutive segment. |
| Delay | vc | ANCOVA | Difference in slope |
| Half-life | HL_FC | *t* test (two-sided, unpaired, unequal variance) | Fold change of mean half-life |
| RNA abundance & half-life | iTSS II & term | MANOVA | Fold change of mean abundance and fold change of mean half-life |
| TI | TI | *t* test (two-sided, unpaired, unequal variance) | Fold change of mean TI termination factor |

segment is adapted to match the bins/probes of the abundance segment before the calculations:

$$R_{synt} = \log 2\left(\frac{\alpha_{3'}}{\alpha_{5'}}\right) = \log 2\left(\frac{int_{3'}}{int_{5'}}\right) + \log 2\left(\frac{\lambda_{3'}}{\lambda_{5'}}\right) = \log 2\left(\frac{int_{3'}}{int_{5'}}\right) - \log 2\left(\frac{HL_{3'}}{HL_{5'}}\right)$$

$R_{synt} > 0$ : new TSS (iTSS_II)

$R_{synt} < 0$ : (partial) termination site (term)

$R_{synt} \approx 0 \,\&\, \log 2\left(\frac{\lambda_{3'}}{\lambda_{5'}}\right) \neq 0$ : processing or stabilization site.

The p-value of $R_{synt}$ is calculated by MANOVA. The independent variable is the segment type (5′ segment = $s_{5'}$ and 3′ segment = $s_{3'}$), while *HL* and RNA abundance are two dependent variables. The partial termination percentage is $t_p = (1 - 2^{R_{synt}}) * 100$, i.e. a $R_{synt} = -0.5$ equals ~29% termination.

Internal TSS identified by a delay drop (iTSS I). The drop is calculated from the respective fitted delays (linear regression) at a position between the two consecutive segments. In order to get the distributions for the t-test, the residuals from the segment-specific linear regressions are summed with the respective fitted delays at the comparison position. To check if two consecutive delay segments have a significantly different elongation rate (velocity event) we checked if the slopes of the respective linear regressions are significantly different with ANCOVA. In this case, segments are the categorical variables, the position is covariate and the delay is the response. We checked for a significant interaction of the categorical variables with the position. For the evaluation of the relation and events between fragments the following statistical tests and measures of differences are considered. For all t-tests and MANOVA we assumed a normal distribution of the values. The p-values for each set of events were adjusted for multiple testing by the method of Benjamini Hochberg. The respective statistical tests for allevents are given in Table 3.

**Statistical assessment of RST segments**. We assessed the probability that the post-rifampicin-increase is due to technical noise. If post-rifampicin-increases are based on technical noise, they should be randomly distributed within all reads/probes. In contrast a systematic effect, such as TI, should affect consecutive reads/probes after an RST event. The background probability of RST bins/probes was calculated from the total number of bins/probes tagged as potential RST bins divided by the total number of bins/probes. In case of *E. coli* BW25113 7420/46803 probes are tagged as potential RST probes resulting in a background probability of 0.159. Based on this background probability we calculated the probability of getting ≥ 0.75 ∗ *n* RST-bins in a sequence of *n* bins, i.e. the probability of getting ≤ 0.25 ∗ *n* non-RST bins, with the cumulative probability function for binomial distributions in R (pbinom) for all segment sizes from 5 to the maximal segment length of an 'rifi' detected RST segment in the respective organisms. The ≥ 0.75 ∗ *n* threshold was used, because by default 'rifi' requires only 75% of RST bins in a potential RST-segment prior to the actual fit (see above). The expected numbers and probabilities of the occurrence of RST segments of length ≥*n* were compared with the actual observed numbers and frequencies (Fig. 4a, Supplementary Fig. 7), showing that the observed RST segments cannot be explained by random sequencing noise. Observed segments needed to fit the following requirements: 1. Segment size ≥ 5 bins; 2. average fitted termination rate ≥0.15; 3. number of individual bins with a fitted termination rate of ≥0.15 needed to be ≥75% of the total number of bins (always rounded up to the next integer to be more stringent).

**RNA-Seq data from published studies**. Rifampicin treatment sequencing data from *Escherichia coli* K12 BW25113 and *Klebsiella aerogenes* KCTC 2190 from the data set of Dar & Sorek[3] was used for the analysis. The data is available at the ENA project ID PRJEB21982. Additionally, we used rifampicin treatment data from *E. coli* K12 MG1655[18] and *Synechococcus* sp. strain PCC 7002[19]. Data available at Gene Expression Omnibus (accession no: GSE75818) or and Gene Expression Omnibus (accession no: GSE109174), respectively. Standard quality control was performed, the reads were aligned to their respective reference genome (*E. coli* BW25113, CP009273.1; *K. aerogenes* KCTC 2190, NC_015663.1; *E. coli* K12 MG1655, NC_000913.2; *Synechococcus* sp. strain PCC 7002, NC_010475.1) using the BWA alignment tool (0.7.17)[44]. Bedtools (2.30.0) genomecov was used to count

the sequencing depth on every position[45]. Spike-in RNAs were used to calculate the normalization factor. Data from *Synechococcus* sp. strain PCC 7002 was readily available as normalized count data. The normalized counts for each base were then averaged into groups of 50 nt, creating bins to reduce the number of samples. The position of the bin refers to the genome position of the last nucleotide in the bin. Samples with an average count below ten were discarded.

***Synechocystis* PCC 6803 microarray data**. Liquid cultures of *Synechocystis* PCC 6803 strain PCC-M were cultivated at 30 °C in standard BG11 medium with constant shaking and illuminated with white light of 50 μmol of photons $m^{-2}s^{-1}$. Triplicate liquid cultures with 250 mL volume were grown at standard conditions until an OD 750 nm of approximately 0.7. Samples for RNA extraction (25 mL each) were taken at time points 0 min (before the addition of rifampicin) and in a time series of 2 min, 4 min, 8 min, 16 min, 32 min and 64 min after the addition of 300 μg/mL rifampicin. For RNA extraction cells were harvested on Supor 800 membranes via vacuum filtration. The folded membrane was transferred into a Sarstedt tube with 1 mL PGTX solution[46], snap frozen in liquid nitrogen and stored at −80 °C until extraction. The RNA extraction process started with 15 min incubation at 65 °C with vortexing several times in between. Then, 700 μL chloroform/IAA (isoamyl alcohol) 24:1 mixture was added to the samples, they were mixed well and incubated for 10 min at room temperature with gentle mixing several times. Afterwards, the samples were centrifuged for 3 min at 6000*g* and the upper phase was transferred to a new tube. The next step was adding 1 volume phenol/chloroform/IAA mixture, centrifuging and transferring the upper phase to the new tube. Next, one volume chloroform/IAA was added, samples were centrifuged and the upper phase was transferred to a new 2 mL RNase free tube. One volume isopropanol was added and mixed to the samples and RNA was precipitated overnight at −20 °C. The next day samples were centrifuged at 4 °C for 12,000*g* for 30 min, the supernatant was removed and the transparent RNA pellet was washed with 200 μL 70% ethanol (without mixing). The samples were centrifuged for 10 min, 12,000*g* at 4 °C, the ethanol was removed, and the RNA pellet was air dried and dissolved in 20 μL RNase free H2O. RNA samples were stored at −80 °C. The RNA samples (10 μg) were DNase treated and precipitated by ethanol overnight, the quality was checked on an agarose gel and concentration was measured by Nanodrop. The RNA was directly labeled[36,47] with the ULS labeling Kit (Kreatech). 2 μg of RNA was mixed with 0.8 μl Cy3-ULS solution per 1 μg of nucleic acid in a volume of 20 μl with 2 μL 10x labeling solution. The samples were incubated at 85 °C for 15 min in the dark and residual unused dye was removed with KREApure columns. The efficiency of labeling was controlled photometrically with the Nanodrop spectrophotometer (microarray mode), which measured simultaneously the absorption at 260 nm (nucleic acid) and 550 nm (Cy3). Agilent custom microarray 8X60k with one color technique were used. The hybridization, washing and scanning procedure was done according to the Agilent protocol for one-color microarray-based gene expression analysis. Shortly, labeled RNA samples were mixed with 10X blocking solution, RNase free water and 25X fragmentation buffer and incubated for 30 min at 60 °C in the dark. Subsequently, 2X Hybridization buffer was carefully mixed with the samples and after short centrifugation they were loaded on the microarray. Hybridization was performed for 17 hs at 65 °C with constant rotation. Subsequent washing was performed with washing buffer 1 and 2 and with acetonitrile for 1 min each. An Agilent Technologies Scanner G2505B was used for the detection of the signals. All array data can be downloaded from NCBI GEO (GSE209879). Signal intensities for probes were obtained from the scanned microarray image using Agilent Technologies' Feature Extraction software version 10.5.1.1 (protocol GE1_105_Dec08). Rifampicin time series data have very asymmetric fold-changes because basically the expression of the whole transcriptome declines compared to the pre rifampicin time point. This is a problem for the common normalization methods such as quantile normalization. As we do not have spike-in RNA samples for an unbiased normalization we resorted to a cyclic loess (cLOESS) based on the least changing dataset, which was done in a similar way for diurnal time series data[48]. We selected genes showing no differential expression when comparing time 0 and after 64 min (log2FC ~ 0). In

practice we used the 1% of probes with the lowest fold change as reference for the cLOESS normalization with limma[49].

**Soft-clustering**. Soft-clustering of the time-series data of individual bins or probes was done with Mfuzz[50] using 20 clusters and a fuzzification parameter of 1.25 on standardized expression data.

**Stochastic simulation**. The simulation was used for an unbiased confirmation of the models for post-transcriptional decay, rifampicin sensitive termination by a cis-asRNA or a fixed termination factor and position as well as the multiple TSS model. Furthermore, it was used to generate an unbiased dataset with known truth for the evaluation of the delay-based 'rifi' estimates (velocity, velocity changes and delay-based TSSs). For the simulation, RNAs are represented as vectors of integers. An RNA ranging e.g. from position 1 to 5 is represented as $r = \{1,2,3,4,5\}$. The integers resemble the positions that are existing in the respective transcript. Growing transcripts and terminated full-length transcripts are stored in separate lists. Potential asRNAs are also stored in dedicated lists. All actions are simulated in 1 s steps, for a predefined number of steps as described in the following.

1. De novo transcription initiation: At each timestep one or more overlapping sense transcripts are initiated with a selectable initiation rate (*initiations/s*) at selectable starting positions ($\geq 1$). A random number between 0 and 1 is generated. If the number is smaller or equal to the respective given initiation rate at the respective position a new vector containing the respective start site as an integer is added to the growing transcript list. Given the starting positions 1 and 100, the respective RNA vectors would be $r = \{1\}$ and $r = \{100\}$.

   a. Rifampicin addition time: If the predefined rifampicin addition time step is reached no further transcription initiations are triggered. All other steps, e.g. elongation or decay continue until the predefined maximum time step is reached.

   b. Initiation of an asRNA: The asRNA is initiated as described for the sense RNA, but written in a dedicated growing asRNA list. In contrast to sense RNAs only one antisense TSS is possible. The start position of the asRNA needs to be given relative to 1, i.e. an asRNA start site of 1000 means that the 5′ end has position 1000 and that the 3′ end extends towards 1 ($r_{as} = \{1000,999,998\}$).

   c. Change of initiation rate: It is possible to set a time point and a set of secondary initiation rates. If the respective time step is reached the secondary rates are taken (only sense transcripts).

2. End of a transcriptional pause: For each RNA in the pause list a random number between 0 and 1 is sampled. If the number is smaller or equal to the pause off-rate, the RNA is moved from the pause list into the growing RNA list.

3. Elongation: Each growing RNA vector is extended by n consecutive ascending integers depending on the defined elongation rate (*nt/s*). Given the RNA vector $r = \{1,2,3,4\}$ and an elongation rate of 5 *nt/s*, the RNA vector after one elongation step is $r = \{1,2,3,4,5,6,7,8,9\}$.

4. Start of a transcriptional pause (only for sense RNAs): If the defined pause start probability is greater than 0, all growing transcripts for which the 3′ end overlaps with the defined pausing site are identified. For each of those RNAs a random number between 0 and 1 is sampled. If the number is smaller or equal to the pause start probability (default: 1) the transcript is moved from the growing RNA list into a pause list. Paused RNAs can still undergo decay but they are excluded from the elongation.

5. Decay: For each existing transcript a random number between 0 and 1 is sampled. If this number is smaller or equal to the input decay constant ($s^{-1}$) the respective transcripts are subjected to the decay function. If transcripts are selected for decay the actual degradation is considered to appear instantaneous within the respective time step.

   a. Post-transcriptional mode: Only full-length RNAs are considered for decay. Selected transcripts are completely deleted.

   b. Co-transcriptional mode: Both growing and full-length transcripts are considered. Selected full-length transcripts are completely deleted. For growing transcripts all positions but the most 3′ prime positions are deleted. The cut transcript can still undergo further rounds of elongation and decay. An example RNA vector $r = \{5,6,7,8,9,10\}$ would become $r = \{10\}$ after the co-transcriptional decay.

   c. Endo_exo mode: This mode simulates endonucleolytic cuts at defined positions and a rapid 3′ to 5′ exonucleolytic decay of the new 3′ ends. For all RNA vectors that contain the predefined endo-site-positions a random number between 0 and 1 is sampled. If this number is smaller or equal to the input decay constant, the rightmost endo site becomes the new 5′ end of the respective RNAs.

6. Termination: If the integer at the 3′ position of a growing RNA vector is equal to or bigger than the selected input total RNA length, the transcript is deleted from the growing RNA list and added to the full-length RNA list. The following additional non-standard termination options exist:

   a. A fixed random termination probability at each time step: For each growing transcript a random number between 0 and 1 is sampled. If this number is equal to or smaller than the given fixed termination probability, the respective transcript is moved to the full-length RNA list before reaching the given maximum transcript length.

   b. Partial termination at given position: For all growing transcripts for which the 3′ position in the growing RNA vector is in a window with the given partial termination site (window size dependent on elongation rate) a random number between 0 and 1 is sampled. If this number is equal to or smaller than the given termination probability, the respective transcript is moved to the full-length RNA list.

   c. Termination by collision: For this mode, the transcription of an cis-asRNA is simulated in parallel. In a first step sense/anti transcript pairs are selected if their 3′ positions are in the same window (default: 30nt). For all selected pairs a random number between 0 and 1 is sampled. If the number is smaller or equal to the given sense collision termination probability (ti_prob_sense) the sense transcript is moved to the full-length sense RNA list. Otherwise the growing asRNA is moved to the full-length asRNA list. A ti_prob_sense <0.5 means that the sense RNA is less often terminated than the asRNA. Collisions without termination are not implemented.

7. Transcript number counting (position dependent): Finally, for a predefined set of positions, the number of transcripts containing the respective position is counted at each time step. Here, all lists (growing RNA list, full-length RNA list, pause list) are considered. Potential asRNAs are counted independently.

**Elongation rate fit accuracy**. To test the accuracy of the velocity estimate a 1000nt transcript with constant elongation rate was simulated. In total 36 parameter combinations (decay constants: 0.003465736, 0.01155, and 0.00578 1/s; initiation rates: 0.2, 0.4 and 0.8 initiations/s) including 4 different elongation rates (15, 30, 45 and 60 nt/s) were used. The simulations for each parameter set were done in triplicates. Poisson noise was added to somewhat resemble technical sequencing noise before fitting and segmenting the data with the penalties used for the Dar & Sorek data (split penalty 3.5, outlier penalty 2.5). This procedure was repeated 50 times for each parameter set. The velocities of the resulting segments were compared with the input velocities.

**Reporting summary**. Further information on research design is available in the Nature Portfolio Reporting Summary linked to this article.

## Data availability

The *Synechocystis* PCC 6803 microarray data from this study have been deposited with the GEO accession number GSE209879. The data sources and respective references for the other organisms are: *Escherichia coli* K12 BW25113 and *Klebsiella aerogenes* KCTC 2190 (Dar and Sorek[3], ENA project ID PRJEB21982). *E. coli* K12 MG1655 (Moffit et al.[18], GEO: GSE75818). *Synechococcus* sp. strain PCC 7002 (Gordon et al.[19], GEO: GSE109174). The 'rifi' R objects containing the source data for all analyses and figures based on experimental data are available in Supplementary Data 5.

## Code availability

The R-package 'rifi', for a holistic analysis of rifampicin data, is available via Bioconductor (https://bioconductor.org/packages/rifi) or GitHub (https://github.com/JensGeorg/rifi), analyses in this paper were done with R-package version 1.2.2 from Bioconductor version 3.16 (https://zenodo.org/record/8054675). The script for stochastic simulation of transcription and degradation at GitHub (https://github.com/JensGeorg/Stochastic-simulation-of-transcription).

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

## Acknowledgements
The authors thank Wolfgang R. Hess for his valuable discussions and insightful comments on the manuscript and Martin Raden for his helpful comments on the dynamic programming strategy. The work was supported by the Deutsche Forschungsgemeinschaft (grant no. DFG GE 3159/1-1) and by the Federal Ministry of Education and Research (BMBF) program RNAProNet (grant no. 031L0164B).

## Author contributions
J.G. designed the study and supervised the research, G.K. performed the *Synechocystis* Experiments and generated the microarray raw data, W.W. and L.Y. did the data pre-processing, W.W., L.Y., and J.G. wrote the 'rifi' code and analyzed the data using 'rifi', J.G. wrote the code for the stochastic simulation and performed/analyzed the simulation experiments, W.W., L.Y., and J.G. wrote and revised the manuscript.

## Funding

## Competing interests
The authors declare no competing interests.
