## [Peer Review File · Communications Biology]

Reviewers' comments:

Reviewer #1 (Remarks to the Author):

The manuscript by Wanney et al. provides insights into the effects of transcriptional interference, which is widespread in all cells, including bacteria. The authors followed up on an unexpected observation of an increase in RNA abundance after exposure to rifampicin (Rif), which is commonly used to stop RNA synthesis and follow the kinetics of RNA decay. Analysis of several Rif datasets in diverse bacteria suggested that this phenomenon is common and led the authors to conclude that it is likely due to Rif-sensitive termination (RST) of convergent transcription. They also provide a platform for the analysis of RNA stability that takes into account the well-established complexity of the bacterial transcriptome, with an RNA segment frequently present in a number of different RNA transcripts that start and stop at different points, invalidating annotation-based analysis for this class of RNAs. While the most complex cases are likely infrequent, this analysis revealed 60+ *E. coli* segments embedded in RNAs that have drastically different half-lives.

I am very familiar with molecular mechanisms and regulation of RNA synthesis, but much less so with genome-wide analysis and definitely not with mathematical modeling. Thus, I can only comment on the mechanistic aspects of this work. I agree that the proposed explanation of the Rif-induced changes in RNA abundance is plausible, and that the authors describe a valuable tool to investigate RNA interference, which is undoubtedly widespread, particularly given that the data are already available (although may lack the required temporal resolution). The ability to accurately determine RNA stability is also very important. Overall, I found the manuscript easy to follow, at least up to Figure 8. The presentation is logical and figures are very helpful in illustrating a rather complex topic. However, a few points could be explained a bit better.

The authors' analysis is consistent with co-transcriptional RNA decay, which I would not have expected to be predominant in *E. coli*, in contrast to *B. subtilis* where 5' exonucleases are implicated in co-transcriptional RNA cleavage. Are there *Bacillus* datasets to compare? And *E. coli* with *ts* alleles of RNaseE?

Rif inhibits RNA polymerase (RNAP) in a non-neutral way: Rif locks the open promoter complex in a reiterative synthesis/release mode. I do not think that an open complex would strongly impede the transcribing RNAP, even though some of them are exceedingly stable, but this is something to keep in mind – the lack of/less pronounced Rif effect could be due to the remaining roadblock.

The authors state that termination would be favored on non-translated RNAs; this is true in some cases but not necessarily in others – in *E. coli*, and presumably in *Klebsiella*, transcription and translation are closely coupled, inhibiting Rho-dependent termination and RNA cleavage. This is not the case in Firmicutes. What do we know about cyanobacteria? If RNAP runs into an obstacle (another RNAP approaching head-on, in this case), there are other ways to release it, and some are not affected by the ribosome (e.g., Mfd).

The authors point out that 14 examples of RST they identified are present in all three Enterobacterial datasets. This is potentially important, and a few of these examples fit the published data. Yet at the same time Fig. 5B shows significant difference between RSTs in MG and BW strains, which are very similar. Perhaps elaborate?

Overall, it appears that *rifi* could be broadly useful to identify cases of TI and to reveal the RNA decay differences hidden in annotation-based analysis. By comparison, I am not convinced that this method could "substitute" for NET analysis of RNAP elongation/pausing. If the estimated rate of RNAP varies between 4 and 170 nt/s within a "single" transcription unit, a better way to arrive at such estimates is needed. Just removing the physiologically irrelevant numbers is not a good solution. There are no accessory factors that can change average RNAP speed more than 3-fold, likely because this would compromise co-transcriptional RNA folding.

Given 50-nt windows, 4 nt/s could reflect a very strong pause, but is there one in *ftsA* or is this a reflection of a second TSS? It is true that RNAP pauses every 100 nt or so, and finding matches in

Larson et al dataset should not be difficult. However, most of these pauses are very short (1 s) and are unlikely to be detectable in Rif timecourses, even the highest resolution ones. Notable exceptions are regulatory pauses in the leader regions of amino acid and LPS core biosynthesis operons; these should be present in both MG and BW strains, and for sure in the waa and wbb leaders (if I recall correctly, E. coli pauses were ranked by Larson et al.). Another possibility for a strong pause would be a chromatin-like structure on DNA. Regardless, I think that the manuscript contains way too much information already, and the authors acknowledge that the "pausing" story is weak. Thus, omitting it altogether would be better, in my opinion.

In table 1, I presume that the termination efficiency is a fraction, not % as shown.

Reviewer #2 (Remarks to the Author):

The paper presents a huge amount of work with a lot of interesting information on transcription/translation dynamics in E.coli.

Overall, I am impressed by the scope of the investigation and recommend publication.

I am also a little confused/overwhelmed by the many data, and the overall parameters deduced from the data. Perhaps it would be good to make some summary of some key take-away points from the analysis.

I.e. the transcription seed and its gene to gene variation? The pause sites, their typical strength in terms of provided delay and their chance to lead to termination.

What is the actual number of transcriptional interference examples that the authors found in E.coli, and how much they in fact did change the expression of the involved gene (their effect would depend on

relative strength of promoters for sense and anti-sense transcripts).

I did not get any clear message from bacteria except E.coli. Why mention these other bacteria without giving some lessons?

Detailed suggestions:

Table 1, I was unable to deduce whether the % termination of e.g. 0.56 meant 0.56% or 56%?

Table 1, last column lacked explanation on what is shown.

I could not find the info about growth medium and temperatures for the reported E.coli measurements. That would be useful when one consider comparison to Moffit et al in Fig. 7A.

Fig. 6B: There is no mark on the x axis that allow reader to estimate estimate the decay times.

half life estimates for 60 can ??? what is 60

Model description starting in page 22 very short, i.e. reader have to guess what parameters alpha and lambda represent. To make paper self consistent it would be good with a figure that describe respective models with key parameters. This may be suitable done together with the description of stochastic simulation.

Stochastic simulation:

- 1) Perhaps reorder list so it follows one mRNA from initiation to elongation to decay.
- 2) Specify that the decay simulation takes place in steps of perhaps 1 sec (?).
- 3) termination with collision, how is it implemented that anti sense transcripts may have different initiation frequency? What does the stochastic choice in point 2c (page 31) actually mean mechanistically?
- 4) Point 3, why at all use time-step where 2 transcription initiations happen simultaneous? It complicates book-keeping, and anyway RNAP's are cannot be initiated faster than about once per second (they in fact have some waiting time in open complex before initiated).

Supplementary fig 1,2, mark minutes(?) on x-axis.

Reviewer #3 (Remarks to the Author):

Time-resolved transcriptome analysis following treatment with rifampicin, an inhibitor of RNA polymerase initiation, has offered a way of studying global RNA dynamics in bacteria. The resulting highly complex data sets have e.g. been used to assess RNA half-life, the mode of RNA degradation, and RNA polymerase elongation rates. Wanney et al. have developed an analysis package ('rifi') for such rifampicin time series. The presented approach is not dependent on a genome annotation – instead covered regions are segmented by observed RNA dynamics (by delay in decay, half-life, and abundance). This segmentation approach allows detecting alternative transcripts and estimating their respective half-lives, whereas annotation-based methods, assuming a single transcript species, would give a single averaged half-life.

Rifampicin time series from different sources were analyzed, across four different bacterial species, dealing with both RNAseq and microarray data sets. Analysis with 'rifi' allowed to identify complex transcript architectures, including internal transcription start sites and partial termination sites. Of particular interest were transcripts that increase in abundance after rifampicin addition ('rifampicin sensitive termination'), an effect the authors linked to anti-sense transcription or trans-acting RNAs. Interestingly, this effect was found for a substantial number of transcripts in all of the investigated data sets, hinting that anti-sense transcription / collisional transcriptional interference is widespread in bacteria. For several of the respective transcripts, the authors were able to relate this effect to previously reported instances of such gene regulation.

The manuscript is clearly written, and the presented analysis method ('rifi') appears to have notable advantages over previous analysis methods, allowing to better investigate transcriptome complexity, dynamics, and regulatory effects from rifampicin series with high temporal resolution.

I have a few comments:

- I think having a graphical overview of the analysis pipeline like a flowchart somewhere (can be in the supplements) might be helpful for readers and potential users of the software.
- In some instances, the authors should be more concise distinguishing collisional transcriptional interference (RNA polymerases colliding) from RNA-mediated effects. For example, the description of collisional transcriptional interference "We expect a much higher termination probability for the untranslated cis-antisense RNAs (asRNAs) upon collision with a translated mRNA" makes it sound as if it describes an RNA interaction, when a collision between RNAPs is meant.

- I was wondering whether the dynamics of rifampicin entering the cells and inhibiting transcription initiation do not need to be considered. I assume this happens on a much faster time scale?
- The figure legend of Figure 1 mentions clustering into 20 clusters. I assume the figure shows time series of those transcripts that fall into clusters characterised by an increase after rifampicin addition. I think it would be good to indicate what respective fraction of total found transcripts they represent, and possibly how many of those 20 clusters are shown in each.
- The Table 1 header indicates termination percentages, but I think the table shows fractions instead of percentages.
- The authors were able to link several of their observed RST instances to previously described gene regulation systems. It could be interesting to also give some predictions / testable hypotheses. For instance, a few (unreported) cases could be pointed out in which significant transcriptional interference is predicted with high confidence but no asRNA can be detected, possibly even narrowing down to the putative anti-sense promoter / promoter region.
- Finally, I noticed a couple of minor formatting / punctuation issues, e.g. instances of incorrectly placed commas, missing spaces after commas, double spaces, occasionally missing a space between number and unit (especially for 'nt'). In the caption of SFigure 17, exponents are not in superscript. Also, for base 10 notations I would recommend using the 'x' character instead of '*' (e.g. 7.6×10^{-9}).

Point by point response

We thank all reviewers for their constructive comments! We have carefully addressed the comments and revised the manuscript accordingly. In addition, we have revised grammar and wording in some instances to further enhance legibility and clarity.

Please note that some numbers that are mostly threshold dependent (e.g. RST instances or “in gene” half life changes), have changed slightly due to new parameters chosen to be consistent across all datasets.

Sincerely

Dr. Jens Georg

Reviewers' comments:

Reviewer #1 (Remarks to the Author):

The manuscript by Wanney et al. provides insights into the effects of transcriptional interference, which is widespread in all cells, including bacteria. The authors followed up on an unexpected observation of an increase in RNA abundance after exposure to rifampicin (Rif), which is commonly used to stop RNA synthesis and follow the kinetics of RNA decay. Analysis of several Rif datasets in diverse bacteria suggested that this phenomenon is common and led the authors to conclude that it is likely due to Rif-sensitive termination (RST) of convergent transcription. They also provide a platform for the analysis of RNA stability that takes into account the well-established complexity of the bacterial transcriptome, with an RNA segment frequently present in a number of different RNA transcripts that start and stop at different points, invalidating annotation-based analysis for this class of RNAs. While the most complex cases are likely infrequent, this analysis revealed 60+ E. coli segments embedded in RNAs that have drastically different half-lives.

I am very familiar with molecular mechanisms and regulation of RNA synthesis, but much less so with genome-wide analysis and definitely not with mathematical modeling. Thus, I can only comment on the mechanistic aspects of this work. I agree that the proposed explanation of the Rif-induced changes in RNA abundance is plausible, and that the authors describe a valuable tool to investigate

RNA interference, which is undoubtedly widespread, particularly given that the data are already available (although may lack the required temporal resolution). The ability to accurately determine RNA stability is also very important. Overall, I found the manuscript easy to follow, at least up to Figure 8. The presentation is logical and figures are very helpful in illustrating a rather complex topic. However, a few points could be explained a bit better.

The authors' analysis is consistent with co-transcriptional RNA decay, which I would not have expected to be predominant in *E. coli*, in contrast to *B. subtilis* where 5' exonucleases are implicated in co-transcriptional RNA cleavage. Are there *Bacillus* datasets to compare? And *E. coli* with ts alleles of RNaseE?

We did not find suitable *Bacillus* or *E. coli* ts RNase E rifampicin datasets with a high temporal and positional resolution. However, *Synechocystis* PCC6803 contains an RNase J homolog which has a 5'-3' exo activity similar to *B. subtilis* RNase J (<https://doi.org/10.3389/fmicb.2020.01055>).

So, yes, it may seem surprising that co-transcriptional RNA cleavage is so prominent. However, the analysis of all investigated datasets from *Enterobacteria* and *Cyanobacteria* indicates mostly co-transcriptional decay. The exact mechanism in *E. coli* remains unclear. A combined RNase E and 3' Exonuclease mechanism seems reasonable. Using the simulation we can see that a very low number of endonucleolytic cuts followed by a rapid 3' exonucleolytic decay shapes the transcriptome towards the co-transcriptional decay pattern. Nevertheless, the recently detected 5'-3' RNase AM (doi.org/10.1021/acs.biochem.5b00192, DOI: 10.1093/nar/gkaa260, doi: 10.1093/nar/gkac617) or other so far undetected 5'-3' RNases might be involved in the decay in *E. coli*, too.

Changes: We extended the discussion about the decay modes in the "Decay modes and decay scenarios" chapter and in the conclusions and included an endonuclease symbol in Fig. 1A.

(page 5, starting at line 151)

"The simplest mechanisms for the co-transcriptional decay would be a 5' exonucleolytic decay as known from B. subtilis. Indeed, Synechocystis PCC6803 and Synechococcus PCC7002 harbor a homolog of the B. subtilis RNase J1 with an 5' exoribonuclease activity^{19,26,27}. However, in E. coli RNA is decayed mainly by the combined action of endo- and 3' exoribonucleases²⁸. In a simulation

example we show that a low number of internal processing sites (3 sites in a 2000 nt transcript) combined with a rapid 3' exonucleolytic decay mimics the co-transcriptional decay by a 5' exoribonuclease (Supplementary Figure 4). Recently, also an 5' exoribonuclease, RNase AM, was discovered in E. coli"

Conclusion:

(page 22, starting at line 626)

"Extending on previous investigations²¹ we show that, if not all, the majority of transcripts from E. coli, K. aerogenes, Synechococcus PCC7002 and Synechocystis PCC6803 can be degraded before the RNAP terminates (co-transcriptional decay). We show that the mode of decay (co-transcriptional vs post-transcriptional) is directly evident from the transcript-wise steady state RNA abundance patterns from standard transcriptomic data without the need of rifampicin time-series - A constant abundance over the full transcript complies with the co-transcriptional decay while the post-transcriptional decay requires a linear abundance decline towards the 3' end. We conclude from stochastic simulation that the co-transcriptional decay patterns can be achieved by 5' exoribonucleases and/or by a combination of endo -and 3' exoribonucleases. Interestingly, Synechocystis PCC6803 and Synechococcus PCC7002 both harbor 5' exo -and endoribonucleases^{19,26,27}, while E. coli is thought to rely mainly on endo- and 3' exoribonucleases²⁸."

We also added the "endo_exo" mode to the simulation description in the methods part (page 35, starting at line 1042)

Rif inhibits RNA polymerase (RNAP) in a non-neutral way: Rif locks the open promoter complex in a reiterative synthesis/release mode. I do not think that an open complex would strongly impede the transcribing RNAP, even though some of them are exceedingly stable, but this is something to keep in mind – the lack of/less pronounced Rif effect could be due to the remaining roadblock.

This is an interesting thought. In order to show an observable effect this rifampicin induced RNAP roadblock needs to survive several collisions with elongating RNAPs. Assuming a very stable roadblock that terminates all elongating RNAPs at e.g an internal promoter, the position directly after the block should show no or only a reduced delay.

This situation can be analyzed using the stochastic simulation. In a 2000 nt transcript with a weaker

internal promoter at 1000 nt ($\frac{1}{3}$ of the primary promoter strength), the weaker promoter does not have a significant effect on the delay if no roadblock is assumed (Fig., left). However, with a rif-induced roadblock a steep drop in the delay is visible (Fig., right). Our 'rifi' workflow would interpret this drop correctly as a new promoter, but might overestimate the internal promoter strength and indicate a new TU start instead of an internal promoter. This could be incorrectly interpreted as termination of the previous transcript. Due to the length of the manuscript we did not include this aspect in the text.

The authors state that termination would be favored on non-translated RNAs; this is true in some cases but not necessarily in others – in *E. coli*, and presumably in *Klebsiella*, transcription and translation are closely coupled, inhibiting Rho-dependent termination and RNA cleavage. This is not the case in Firmicutes. What do we know about cyanobacteria? If RNAP runs into an obstacle (another RNAP approaching head-on, in this case), there are other ways to release it, and some are not affected by the ribosome (e.g., Mfd).

We specified the “termination of untranslated RNAs” statement and wrote that it is reported for *E. coli*. For Cyanobacteria to our knowledge nothing is known about the stability of translated vs. untranslated mRNAs. We have added two examples from PCC6803 for a positional abundance decline that do not comply with the post-transcriptional decay to Supplementary Figure 3.

We also briefly discuss the implications of coupled- and uncoupled transcription-translation in the conclusion:

(page 23, starting at line 636)

*“The co-transcriptional decay is consistent with coupled transcription-translation, which is common in many prokaryotes⁴². However, uncoupled transcription-translation is also frequent. In *B. subtilis* the transcription elongation outpaces the translation which leads to an uncoupling⁴³ and there are also many examples for spatiotemporal separation of transcription and translation⁴². Nevertheless, the contradiction of co-transcriptional decay and uncoupled transcription-translation is not as big as it may be expected. As mentioned before²¹, the co-transcriptional decay is a stochastic process and depending on the decay constant, the length of the transcript and the elongation rate a given percentage of full-length transcript can be expected in the steady state. For a transcript length of 1000 nt, a half-life of 2 min and an elongation rate of 25 nt/s roughly 79% of transcripts are not degraded before the termination (Supplementary Figure 21). “*

The authors point out that 14 examples of RST they identified are present in all three Enterobacterial datasets. This is potentially important, and a few of these examples fit the published data. Yet at the same time Fig. 5B shows significant difference between RSTs in MG and BW strains, which are very similar. Perhaps elaborate?

The differences in the RST numbers (Fig. 5B) are most likely due to the different experimental setups, especially the time-resolutions used. This aspect is currently discussed at the end of the RST passage: *“The temporal resolution is a critical aspect for RST detection. This is reflected in the lower number of RST instances for the datasets of *E. coli* MG1655 and *Synechocystis* PCC6803 (Fig. 5B), which both have a lower temporal resolution and lack e.g. a 1 min sampling point”*. We added a sentence directly after the first reference to Fig. 5B.

New sentence: (page 10, starting at line 269) *“The significant differences between the RST numbers in the investigated organism are most likely due to the different time-resolutions in the respective experimental setups as discussed below.”*

Overall, it appears that *rifi* could be broadly useful to identify cases of TI and to reveal the RNA decay differences hidden in annotation-based analysis. By comparison, I am not convinced that this method could “substitute” for NET analysis of RNAP elongation/pausing. If the estimated rate of RNAP varies between 4 and 170 nt/s within a “single” transcription unit, a better way to arrive at such estimates is needed. Just removing the physiologically irrelevant numbers is not a good solution. There are no

accessory factors that can change average RNAP speed more than 3-fold, likely because this would compromise co-transcriptional RNA folding.

We totally agree that all delay-based estimates such as the elongation rate and the pausing site detection are rather inaccurate. This is due to the usually low time-resolution of rifampicin datasets, resulting in a very low number of datapoints that can be used for the delay fit. This general problem is already briefly discussed in the main manuscript: *“For all delay-based estimates it needs to be considered, that the delay coefficient fit relies only on a low number of datapoints, due to the relatively low temporal resolution of typical rifampicin data. This is e.g. reflected in the tendency of the delay to be fitted at distinct points from the time series, as exemplarily shown for E. coli BW25113 (Supplementary Fig. 15A).”*

We moved the above sentence to an earlier position to be more prominent. (now: page 18, starting at line 504)

Elongation rate: Due to lack of the knowledge about the biological truth, we tried to approximate the accuracy of the elongation-rate-estimate using stochastic simulation. We added Fig. 8D with a simulation based elongation rate accuracy test and the following sentence to the main manuscript:

(page 18, starting at line 507)

“Chen et al. used the delay to calculate the transcription elongation rate. We tried to estimate the expected accuracies for the elongation rates based on simulated data. The calculated velocities are clearly correlated with the actual input velocities, but show a significant dispersion (Fig. 8D). In conclusion, with the given time-resolution, the delay based elongation rates for individual transcripts are rather rough estimates.”

Given 50-nt windows, 4 nt/s could reflect a very strong pause, but is there one in *ftsA* or is this a reflection of a second TSS?

Yes, another aspect is the problem with “overlapping” transcription”, i.e. that one position is covered by transcripts that have different transcriptional start sites. In those cases the simple fitting model is violated and the delay is incorrect. This applies specifically to the 4nt/s example in the *ftsA* gene. We tried to discuss this problem: *“Actually, there are many iTSS known within operons and genes. This leads to overlapping transcription processes and transcripts initiated from different starting sites, which violates the simple delayed-co-transcriptional decay model. Fitting those cases with the simple model can lead to an “artificial” delay (Supplementary Fig. 14B), which lies between the delay of the two independent transcription processes.”*

We modified Fig. 8 (see above) and added the sentence:

(page 19, starting at line 534)

*“Nevertheless, it needs to be considered that the “artificial” delay can also lead to an incorrect elongation rate calculation. An example is the very low calculated elongation rate of 4nt/s for the transcript segment after the iTSS in *ftsA* (Fig. 8A).”*

It is true that RNAP pauses every 100 nt or so, and finding matches in Larson et al dataset should not be difficult. However, most of these pauses are very short (1 s) and are unlikely to be detectable in Rif timecourses, even the highest resolution ones. Notable exceptions are regulatory pauses in the leader regions of amino acid and LPS core biosynthesis operons; these should be present in both MG and BW strains, and for sure in the waa and wbb leaders (if I recall correctly, E. coli pauses were ranked by Larson et al.). Another possibility for a strong pause would be a chromatin-like structure on DNA. Regardless, I think that the manuscript contains way too much information already, and the authors acknowledge that the “pausing” story is weak. Thus, omitting it altogether would be better, in my opinion.

We found it fascinating that “longer” pausing sites might be detectable by jumps in the delay. However, we agree that the pausing story is weak and omitted it from the manuscript.

In table 1, I presume that the termination efficiency is a fraction, not % as shown.

Yes, the table showed the fraction instead of the %. We fixed this problem.

Reviewer #2 (Remarks to the Author):

The paper presents a huge amount of work with a lot of interesting information on transcription/translation dynamics in E.coli.

Overall, I am impressed by the scope of the investigation and recommend publication.

I am also a little confused/overwhelmed by the many data, and the overall parameters deduced from the data. Perhaps it would be good to make some summary of some key take-away points from the analysis. I.e. the transcription seed and its gene to gene variation? The pause sites, their typical strength in terms of provided delay and their chance to lead to termination.

Especially pausing sites and elongation rates (delay based estimates) are not that reliable. We discussed this issue in greater detail in the manuscript now and removed the “pausing” part as suggested by reviewer 1. We tried to summarize all key points in a paragraph at the end of the introduction (page 3, starting at line 84) and in the conclusion section.

What is the actual number of transcriptional interference examples that the authors found in *E. coli*, and how much they in fact did change the expression of the involved gene (their effect would depend on relative strength of promoters for sense and anti-sense transcripts).

The total numbers depend on the selection stringency (compare Fig. 5B). We added the actual numbers to the barplots in the figure. In *E. coli* BW25113 at least 39 of the events are associated with a termination event/ synthesis rate drop, indicating a significant effect on the involved genes. We add the information from *E. coli* BW25113 with the respective termination rate as a supplemental file. In case of collision TI, the effects are indeed dependent on the relative promoter strengths. However, if we do not see coverage for an asRNA we can not exclude RST by e.g. *trans*-acting sRNAs.

I did not get any clear message from bacteria except *E. coli*. Why mention these other bacteria without giving some lessons?

The data from the other bacteria are important to show several aspects:

1. RST, probable by TI and sRNAs, is not limited to a specific dataset. Some RST events are even conserved within the enterobacterial datasets. Beyond the biological implications this excludes technical biases.
2. The rfi tool works for very diverse bacterial groups (*Cyanobacteria*, *Proteobacteria*) and different experimental platforms (RNA-Seq, microarrays).
3. We could show that the co-transcriptional decay case is pre-dominant beyond the diverse bacterial groups, despite the different RNase equipment.
4. We focussed for the main examples on *E. coli* due to the best literature coverage but we used examples from all 4 organisms to depict specific findings.

We agree that the above aspects were not well described in the manuscript. For that reason we extended the discussion about the other organism at several positions and added a table (Table 2) that compares the results from all organisms.

Detailed suggestions:

Table 1, I was unable to deduce whether the % termination of e.g. 0.56 meant 0.56% or 56%?

We used the fraction instead of the percentage. 0.56 means actually 56%. We corrected the table.

Table 1, last column lacked explanation on what is shown.

We added the following information to the table legend: "*The final column shows if the RST event is conserved between the two E. coli strains and/or K. aerogenes.*" We also changed the name of the column to "*RST event conserved in:*"

I could not find the info about growth medium and temperatures for the reported *E. coli* measurements. That would be useful when one consider comparison to Moffit et al in Fig. 7A.

Both half-life estimates in Fig. 7A are based on the same raw data from the Moffit *et al* study. The only difference is the analysis method (annotation-based (original study) vs. not annotation based (this study). We added the comment "*raw data from Moffit et al*" at the y-axis and added the following sentence to the figure legend: "*The plot compares the results from the original, annotation based*

analysis, from Moffit et al. (x-axis) with the annotation independent workflow from this study based on the same raw data”.

Furthermore we added a table (Table2), which compares the data from all organisms and the respective key experimental parameters, to the conclusion.

Fig. 6B: There is no mark on the x axis that allow reader to estimate estimate the decay times.

We assume Fig. 7B was meant, we added ticks on the x-axis for the plots in Fig. 7.

half life estimates for 60 can ??? what is 60

We added the missing word “genes”.

Model description starting in page 22 very short, i.e. reader have to guess what parameters alpha and lambda represent. To make paper self consistent it would be good with a figure that describe respective models with key parameters. This may be suitable done together with the description of stochastic simulation.

We defined all used parameters directly at the start of the model section or when they are first used.

We did not repeat the definition before every model. We added a sentence to summarize the parameters at the start of the section:

(page 25, starting at line 708)

“Used parameters: λ = decay constant [1/s], α = synthesis rate [mol/s], v = elongation rate [nt/s], n = position relativ to the TSS [nt], t = time after rifampicin addition [s], L = total transcript length [nt], n_{term} = position of rifampicin sensitive termination [nt], β = termination proportion.”

We also added Supplementary Figure 22 that graphically explains the models.

Stochastic simulation:

1) Perhaps reorder list so it follows one mRNA from initiation to elongation to decay.

We re-ordered the description of the simulation for a more logical structure. For consistency we also re-ordered the script on Github. This does not change any results.

2) Specify that the decay simulation takes place in steps of perhaps 1 sec (?).

The description of the simulation contains the sentence: *“All actions are simulated in 1 second steps, for a predefined number of steps as described in the following.”*

3) termination with collision, how is it implemented that anti sense transcripts may have different initiation frequency?

The initiation frequenc(y)ies for individual sense RNA TSSs (`pol_freq`) and asRNAs TSSs (`ti_anti_pol_freq`) are defined by different input parameters. The initiation and elongation of sense and asRNAs is done in different code chunks and the “RNA vectors” are stored in independent lists.

For more than one sense or asRNA promoter a vector of initiation rates and start sites needs to be defined. The script loops over all start sites.

What does the stochastic choice in point 2c (page 31) actually means mechanistically?

The script allows defining different probabilities for the termination of sense and antisense transcription upon a collision with the “`ti_prob_sense`” parameter. `ti_prob_sense = 0.5` would mean that sense and antisense termination appear equally often. We usually use a `ti_prob_sense` below 0.5, i.e. a lower termination probability for the sense RNA, because it was reported for E. coli that the translated sense RNA is less often terminated than the un-translated asRNA. We modified the description to be more clear:

(page 36, starting at line 1063)

“If the number is smaller or equal to the given sense collision termination probability (ti_prob_sense) the sense transcript is moved to the full-length sense RNA list. Otherwise the growing asRNA is moved to the full-length asRNA list. A $ti_prob_sense < 0.5$ means that the sense RNA is less often terminated than the asRNA.”

4) Point 3, why at all use time-step where 2 transcription initiations happen simultaneous? It complicates book-keeping, and anyway RNAP's are cannot be initiated faster than about once per second (they in fact have some waiting time in open complex before initiated).

Cyanobacteria can be polyploid with partly >100 genome copies. We implemented this feature to be able to simulate more than one transcription process in one cell.

Supplementary fig 1,2, mark minutes(?) on x-axis.

We added the respective unit (min) to the plots.

Reviewer #3 (Remarks to the Author):

Time-resolved transcriptome analysis following treatment with rifampicin, an inhibitor of RNA polymerase initiation, has offered a way of studying global RNA dynamics in bacteria. The resulting highly complex data sets have e.g. been used to assess RNA half-life, the mode of RNA degradation, and RNA polymerase elongation rates. Wanney et al. have developed an analysis package ('rifi') for such rifampicin time series. The presented approach is not dependent on a genome annotation – instead covered regions are segmented by observed RNA dynamics (by delay in decay, half-life, and abundance). This segmentation approach allows detecting alternative transcripts and estimating their respective half-lives, whereas annotation-based methods, assuming a single transcript species, would give a single averaged half-life.

Rifampicin time series from different sources were analyzed, across four different bacterial species, dealing with both RNAseq and microarray data sets. Analysis with 'rifi' allowed to identify complex transcript architectures, including internal transcription start sites and partial termination sites. Of particular interest were transcripts that increase in abundance after rifampicin addition ('rifampicin sensitive termination'), an effect the authors linked to anti-sense transcription or trans-acting RNAs.

Interestingly, this effect was found for a substantial number of transcripts in all of the investigated data sets, hinting that anti-sense transcription / collisional transcriptional interference is widespread in bacteria. For several of the respective transcripts, the authors were able to relate this effect to previously reported instances of such gene regulation.

The manuscript is clearly written, and the presented analysis method ('rifi') appears to have notable advantages over previous analysis methods, allowing to better investigate transcriptome complexity, dynamics, and regulatory effects from rifampicin series with high temporal resolution.

I have a few comments:

- I think having a graphical overview of the analysis pipeline like a flowchart somewhere (can be in the supplements) might be helpful for readers and potential users of the software.

We added a graphical flowchart overview of the 'rifi' package to the supplement (Supplementary Fig. 14).

- In some instances, the authors should be more concise distinguishing collisional transcriptional interference (RNA polymerases colliding) from RNA-mediated effects. For example, the description of collisional transcriptional interference “We expect a much higher termination probability for the un-translated cis-antisense RNAs (asRNAs) upon collision with a translated mRNA” makes it sound as if it describes an RNA interaction, when a collision between RNAPs is meant.

We specified that the collision of RNAPs is meant. (page 7, starting at line 205)

“We expect a much higher termination probability for the RNAP transcribing the un-translated cis-antisense RNAs (asRNAs) upon collision with the RNAP transcribing a translated mRNA”

- I was wondering whether the dynamics of rifampicin entering the cells and inhibiting transcription initiation do not need to be considered. I assume this happens on a much faster time scale?

It is actually hard to find data about the actual uptake dynamics of rifampicin for diverse bacteria in the literature. We added this information to the methods section of the manuscript.

(page 26, starting at line 747)

*“The lag of the action of rifampicin after addition to the media is between 5 to 10 seconds in *E. coli*²¹. Assuming that all RNAPs are equally affected by the lag, this should result in an ubiquitous delay of some seconds for the 5' ends of all transcripts. For hypothetical data with a much higher temporal resolution in the time frame <30s this ubiquitous delay would be detectable by the delayed co-transcriptional model. In reality the temporal resolution is much lower and also an experimental error of some seconds in the time taking for the sample time points can be assumed. In conclusion the small lag of the rifampicin action should not seriously affect the results.”*

The figure legend of Figure 1 mentions clustering into 20 clusters. I assume the figure shows time series of those transcripts that fall into clusters characterised by an increase after rifampicin addition. I think it would be good to indicate what respective fraction of total found transcripts they represent, and possibly how many of those 20 clusters are shown in each.

Actually, the cluster section was not well described. We did not use transcript based time-series data for the clustering, but data from individual probes (microarray) or from individual bins (RNA-seq). Each presented sub-figure shows one representative cluster for each organism that shows a post rifampicin abundance increase. We improved the description in the main text, the figure legend and the method section.

We did not include numbers here because we used the data only for a qualitative analysis at this point. Nevertheless, it is interesting what fraction of the respective probes/bins show an abundance increase in each dataset. In order to identify candidate bins we did not use the soft clustering, but filtered by an increase of at least 7.5% respective to t=0 in a later time point. The “increase-fraction”

based on this analysis is given for *E. coli* BW25113 in Fig. 5A (0.159) and for the other organisms in Supplementary Figure 5.

- The Table 1 header indicates termination percentages, but I think the table shows fractions instead of percentages.

We corrected the table.

- The authors were able to link several of their observed RST instances to previously described gene regulation systems. It could be interesting to also give some predictions / testable hypotheses. For instance, a few (unreported) cases could be pointed out in which significant transcriptional interference is predicted with high confidence but no asRNA can be detected, possibly even narrowing down to the putative anti-sense promoter / promoter region.

There is a plethora of such candidates which makes it hard to choose. Top candidates for *E. coli* can be e.g. found in Table 1 or Supplementary Dataset 2. We now additionally added an Supplementary table with all top RST candidates from the 5 datasets.

- Finally, I noticed a couple of minor formatting / punctuation issues, e.g. instances of incorrectly placed commas, missing spaces after commas, double spaces, occasionally missing a space between number and unit (especially for 'nt'). In the caption of SFigure 17, exponents are not in superscript. Also, for base 10 notations I would recommend using the '×' character instead of '**' (e.g. 7.6×10^{-9}).

We tried to spot and correct these errors.

Reviewer #1 (Remarks to the Author):

This manuscript reports significant and interesting findings on the complexity and regulation of RNA synthesis and degradation in bacteria. Importantly, this work also provides a new and broadly useful tool for the analyses of transcriptomics data by other groups.

In the revised manuscript, the authors have addressed all my concerns. I have only a few minor corrections:

Line 66 delete comma after "Chen et al..."

Line 91 "termination efficiency" instead of percentage

Line 504 comma not needed in "...considered, that..."

Line 626 add comma after "Extending on previous studies..."

Reviewer #2 (Remarks to the Author):

I am happy with the changes and have no further suggestions. I recommend publication.

Reviewer #3 (Remarks to the Author):

The authors have comprehensively addressed reviewers' comments raised in the first round of review, and the manuscript has been improved in clarity.